# Single-cell profiling reveals distinct adaptive immune hallmarks in MDA5+ dermatomyositis with therapeutic implications

Yan Ye[1,6], Zechuan Chen[2,3,6], Shan Jiang[2,3,6], Fengyun Jia[2,3], Teng Li[2,3], Xia Lu[1], Jing Xue[4], Xinyue Lian[1], Jiaqiang Ma[2], Pei Hao[2,3], Liangjing Lu[1], Shuang Ye[1], Nan Shen[1], Chunde Bao[1], Qiong Fu[1] ✉ & Xiaoming Zhang[2,3,5] ✉

Anti-melanoma differentiation-associated gene 5-positive dermatomyositis (MDA5+ DM) is an autoimmune condition associated with rapidly progressive interstitial lung disease and high mortality. The aetiology and pathogenesis of MDA5+ DM are still largely unknown. Here we describe the immune signatures of MDA5+ DM via single-cell RNA sequencing, flow cytometry and multiplex immunohistochemistry in peripheral B and T cells and in affected lung tissue samples from one patient. We find strong peripheral antibody-secreting cell and CD8+ T cell responses as cellular immune hallmarks, and over-stimulated type I interferon signaling and associated metabolic reprogramming as molecular immune signature in MDA5+ DM. High frequency of circulating ISG15+ CD8+ T cells at baseline predicts poor one-year survival in MDA5+ DM patients. In affected lungs, we find profuse immune cells infiltration, which likely contributes to the pro-fibrotic response via type I interferon production. The importance of type I interferons in MDA5+ DM pathology is further emphasized by our observation in a retrospective cohort of MDA5+ DM patients that combined calcineurin and Janus kinase inhibitor therapy show superior efficacy to calcineurin inhibitor monotherapy. In summary, this study reveals key immune-pathogenic features of MDA5+ DM and provides a potential basis for future tailored therapies.

Anti-melanoma differentiation-associated gene 5-positive dermato-myositis (MDA5+ DM) is an infrequent but distinct subtype of idio-pathic inflammatory myopathies (IIM). Patients with MDA5+ DM are characterized by anti-MDA5 autoantibody, and usually present with the typical skin eruption of DM but few or no features of clinical muscle weakness[1,2]. Interstitial lung disease (ILD) is the most common com-plication in MDA5+ DM. In East Asia, especially in Japan and China, ~40% of MDA5+ DM patients develop life-threatening rapidly-progressive

interstitial lung disease (RP-ILD)[3,4]. In recent years, MDA5+ DM has begun to be recognized in Western countries. In a report from French Myositis Network, around 20% of MDA5+ DM patients develop RP-ILD with a mortality rate of more than 80%[5].

The aetiology and pathogenesis of MDA5+ DM remain elusive. Interestingly, striking similarities in clinical manifestations such as lung pathogenic features, have been observed between MDA5+ DM and severe Coronavirus Disease 2019 (COVID-19)[6]. Furthermore, anti-

[1]Department of Rheumatology, Renji Hospital, Shanghai Jiaotong University School of Medicine, Shanghai 200001, China. [2]The Center for Microbes, Development and Health, Key Laboratory of Molecular Virology and Immunology, Institut Pasteur of Shanghai, Chinese Academy of Sciences, Shanghai 200031, China. [3]University of Chinese Academy of Sciences, Beijing 100049, China. [4]Department of Rheumatology, the Second Affiliated Hospital of Zhejiang University School of Medicine, Hangzhou 310009, China. [5]Shanghai Huashen Institute of Microbes and Infections, Shanghai 200052, China. [6]These authors contributed equally: Yan Ye, Zechuan Chen, Shan Jiang. ✉e-mail: fuqiong@renji.com; xmzhang@ips.ac.cn

MDA5 autoantibody can be detected in the sera of COVID-19 patients whose titers were positively associated with disease severity[7]. Thus, great interests have been raised to explore MDA5[+] DM in the fields of rheumatology and immunology, given rapid research progress in the study of COVID-19.

Due to the lack of specific targets, current treatments for MDA5[+] DM are largely empirical. Immunosuppressive treatments, including triple therapy (high-dose glucocorticoids, tacrolimus, and intravenous cyclophosphamide)[8] and Janus kinase inhibitor-based therapy[9], are currently the mainstream regimens for MDA5[+] DM, which could significantly improve the survival of early-stage patients. However, the overall mortality is still strikingly high in advanced-stage patients who do not respond to conventional immunosuppressive therapies, with one-year mortality rate as high as 50%[4,10]. Thus, within the category of IIMs, MDA5[+] DM represents the most severe subtype and poses a remarkable clinical challenge.

Several lines of evidence indicate that a dysregulated autoantibody response is strongly involved in the disease progression of MDA5[+] DM. The key feature of this disease is the generation of anti-MDA5 autoantibody first described by Sato et al. in 2005[11]. Anti-MDA5 antibody has become the gold standard to diagnose MDA5[+] DM. In addition, anti-MDA5 titers are dynamic, decreased in remission and resurged in disease flares[12], and sustained high levels of anti-MDA5 are associated with unfavorable outcomes in those patients refractory to treatment[10,13]. Recent studies have found that MDA5[+] DM patients with co-existing anti-Ro52 antibodies have an increased frequency of RP-ILD and more aggressive phenotypes[14,15], further highlighting that the breach of B cell tolerance likely contributes to the pathogenesis in MDA5[+] DM.

T cell abnormalities are also reported in MDA5[+] DM. Patients show a decrease in peripheral blood CD4[+] and CD8[+] T cell counts, and CD8[+] T cells decreased more significantly than CD4[+] T cells in patients with deterioration of ILD[16,17]. In another study, the frequency of CD4[+]CXCR4[+] T cells was increased in the blood, and bronchoalveolar lavage fluid of MDA5[+] DM patients, and these T cells can promote pulmonary fibroblast proliferation via interleukin-21 (IL-21), suggesting a potential pathogenic role in MDA5[+] DM[18]. The inclusion of calcineurin inhibitors (CNIs; cyclosporine A, tacrolimus) in several immunosuppressive regimens indicates that targeting T cells is an effective means to treat MDA5[+] DM[4,19].

Here we explore the adaptive immune landscape of MDA5[+] DM by comprehensive single-cell studies, including single-cell RNA sequencing (scRNA-seq) and B/T cell receptor sequencing (scBCR/TCR-seq) in MDA5[+] DM patients, as well as IIM disease and healthy controls. We identify strong activations of antibody-secreting cell and CD8[+] T cell responses, and the underlying type I interferon signaling as key cellular and molecular immune features of MDA5[+] DM, respectively. Furthermore, co-inhibiting T cell activation and Janus kinase pathway shows promising therapeutic efficacy in MDA5[+] DM patients. Thus, the current study provides new clues for the immunopathogenesis and potential therapeutic targets for MDA5[+] DM.

## Results

### Single-cell landscape of peripheral B and T cells from MDA5[+] DM and control groups

To explore the adaptive immune responses in MDA5[+] DM patients at the single-cell level, we performed scRNA-seq on peripheral B and T cells from seven active MDA5[+] DM patients (MDA5[+] DM-Act), three paired MDA5[+] DM patients in remission (MDA5[+] DM-Rem), and five IIM disease controls (Ctrl IIM) and four healthy donors (HD) (Fig. 1a, Supplementary Fig. 1a, b and Supplementary Table 1). We also performed scRNA-seq on lung tissue from one end-stage MDA5[+] DM patient who underwent lung transplantation (Fig. 1a and Supplementary Table 1). After stringent computational quality control, 111,200 high-quality B and T cells from peripheral blood were obtained, with an average of 5674 unique molecular identifiers, representing an average

of 1587 genes (Supplementary Fig. 1c). No obvious batch effect was observed across different samples, and these cells were integrated for graph-based clustering. Cell clusters were defined by established cell marker genes (Supplementary Fig. 1d and Supplementary Table 2). Using t-distributed stochastic neighbor embedding (t-SNE) projection, peripheral B and T cells were classified into 26 clusters, namely eight B cell clusters, nine CD4[+] T cell clusters, seven CD8[+] T cell clusters, and two unconventional T cell clusters (Fig. 1b).

Notable differences were observed between the four groups, as illustrated by t-SNE projection (Fig. 1c) and the fraction of each cluster among the B and T cells (Supplementary Fig. 1e). We also performed clustering analysis with TooManyCells[20] and found that the MDA5[+] DM-Act group was overrepresented by two large branches while the three other groups were well separated as small branches (Fig. 1d). The group preference of each cluster was further illustrated based on the ratio of observed to randomly expected cell numbers used to remove the technical variations in tissue preference estimation (Fig. 1e)[21]. In particular, two antibody-secreting cell (ASC) clusters, interferon-stimulating gene[+] (ISG[+]) B cell, and CD4[+] and CD8[+] T cell clusters were more enriched in the MDA5[+] DM-Act group than in the three other groups (Fig. 1e). These results suggest a uniquely activated status of the adaptive immune system in MDA5[+] DM-Act patients.

### Exaggerated ASC response in MDA5[+] DM patients

Next, we evaluated the peripheral B cell compartment in details across the four groups. Among the eight defined B cell clusters by scRNA-seq, three belonged to naïve B cells (scB1-Transitional, scB2-ISG, scB3-Naïve), three were memory B cells (Bm) (scB4-Unswitched Bm, scB5-Switch Bm [switched Bm], scB6-atypical Bm), and two were ASCs (scB7-pASC [proliferating ASC] and scB8-rASC [resting ASC]) (Fig. 2a). Compared with the Ctrl IIM and HD groups, the MDA5[+] DM-Act group showed significantly higher proportions of scB2-ISG, scB7-pASC, and scB8-rASC cells (Fig. 2b and Supplementary Fig. 2a, b). Furthermore, the proportions of the three clusters were strongly decreased in the three paired MDA5[+] DM-Rem patients (Fig. 2b). Next, we used flow cytometry to validate this finding in an independent cohort (Supplementary Table 3). The frequencies of CD19[+]CD20[-]CD38[hi] ASCs were significantly higher in MDA5[+] DM-Act patients ($n = 36$) than in Ctrl IIM patients ($n = 20$) and HDs ($n = 16$) (Fig. 2c and Supplementary Fig. 2c). In eight paired MDA5[+] DM patients, the frequencies of ASCs were significantly decreased after remission (Fig. 2c). These results reveal that the B cell compartment is greatly activated with the terminal differentiation of ASCs in active MDA5[+] DM patients.

We further evaluated the B cell response by analyzing BCR repertoires via scBCR-seq. Clonotype analysis indicated that clonal expansions were mainly restricted to ASCs (Fig. 2d). We then focused on the analysis of ASCs across the four groups. The MDA5[+] DM-Act group showed a much higher top 20 clonotype frequency of ASCs than those in other groups, indicating that high clonal expansion occurred in the MDA5[+] DM-Act patients (Fig. 2e). We further analyzed BCR clonal sharing between the terminal scB8-rASC and other B cell clusters using the STARTRAC algorithm[21]. The results showed that scB7-pASC, and to a lesser extent, scB5-Switch Bm were clonally connected with scB8-rASC (Fig. 2f), suggesting a possible differentiation link between switched Bm, pASCs, and rASCs.

To explore the immunoglobulin heavy chain variable region (IGHV) gene usages of ASCs in MDA5[+] DM-Act patients, we calculated the fold changes of IGHV genes between MDA5[+] DM-Act, HD, and Ctrl IIM groups. We observed that a panel of IGHV genes were uniquely increased in the MDA5[+] DM-Act group, namely IGHV1-2, IGHV1-3, IGHV1-46, IGHV1-69-2, IGHV2-5, IGHV3-9, IGHV4-31, IGHV4-34, and IGHV7-4-1 (Fig. 2g and Supplementary Fig. 2d). Of note, many of the increased IGHV genes in the MDA5[+] DM-Act patients are implicated in several autoimmune diseases. An increase in the IGHV1 family (IGHV1-2, IGHV1-3, IGHV1-46, IGHV1-69), and IGHV4-4 is strongly associated with

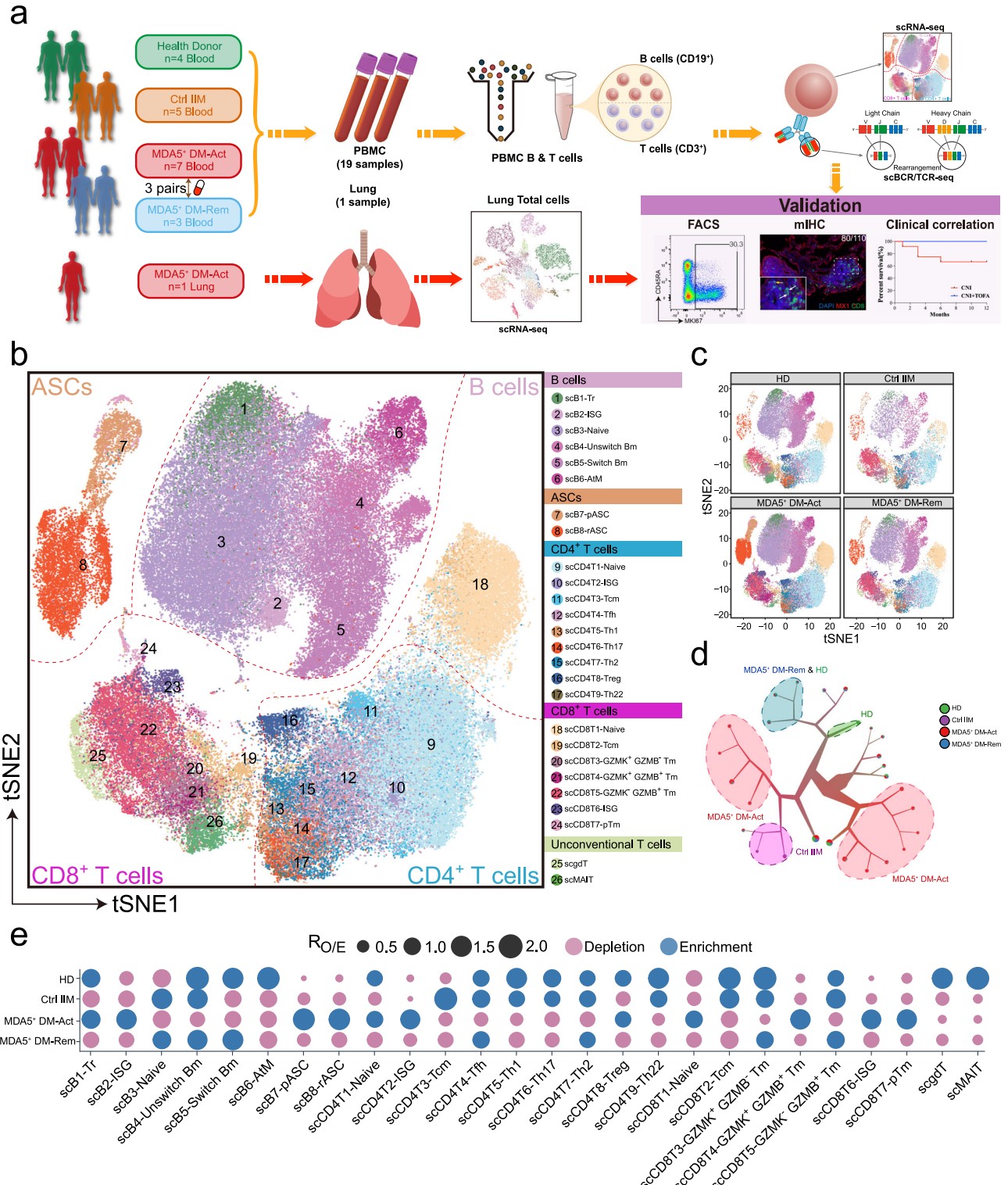

**Fig. 1 | Single-cell atlas of peripheral B and T cells from MDA5+ DM patients and controls. a** Flowchart depicting the overall experimental design of this study. Adobe Illustrator 2020 was used to create the image. **b** t-SNE plot showing the overview of 26 cell clusters in the integrated single-cell transcriptomes of 111,200 peripheral B and T cells derived from MDA5+ DM patients, IIM disease controls, and HDs (Supplementary Table 1). Clusters are named according to the indicated immune cell subsets using the specific gene expression patterns (Supplementary Table 2) and are color-coded. **c** t-SNE plots showing the single-cell transcriptomes

of peripheral B and T cells from HD, Ctrl IIM, MDA5+ DM-Act, and MDA5+ DM-Rem groups. Cell subsets are color-coded as in (**b**). **d** TooManyCells clustering tree to display the scRNA-seq data as cell clades for the four groups. Cells are divided into clusters and related clusters rather than being presented as individual cells. **e** Patient group preference of each cluster measured by the ratio of observed to randomly expected cell numbers (R_{O/E}) calculated by the STARTRAC-dist alogrithm[21].

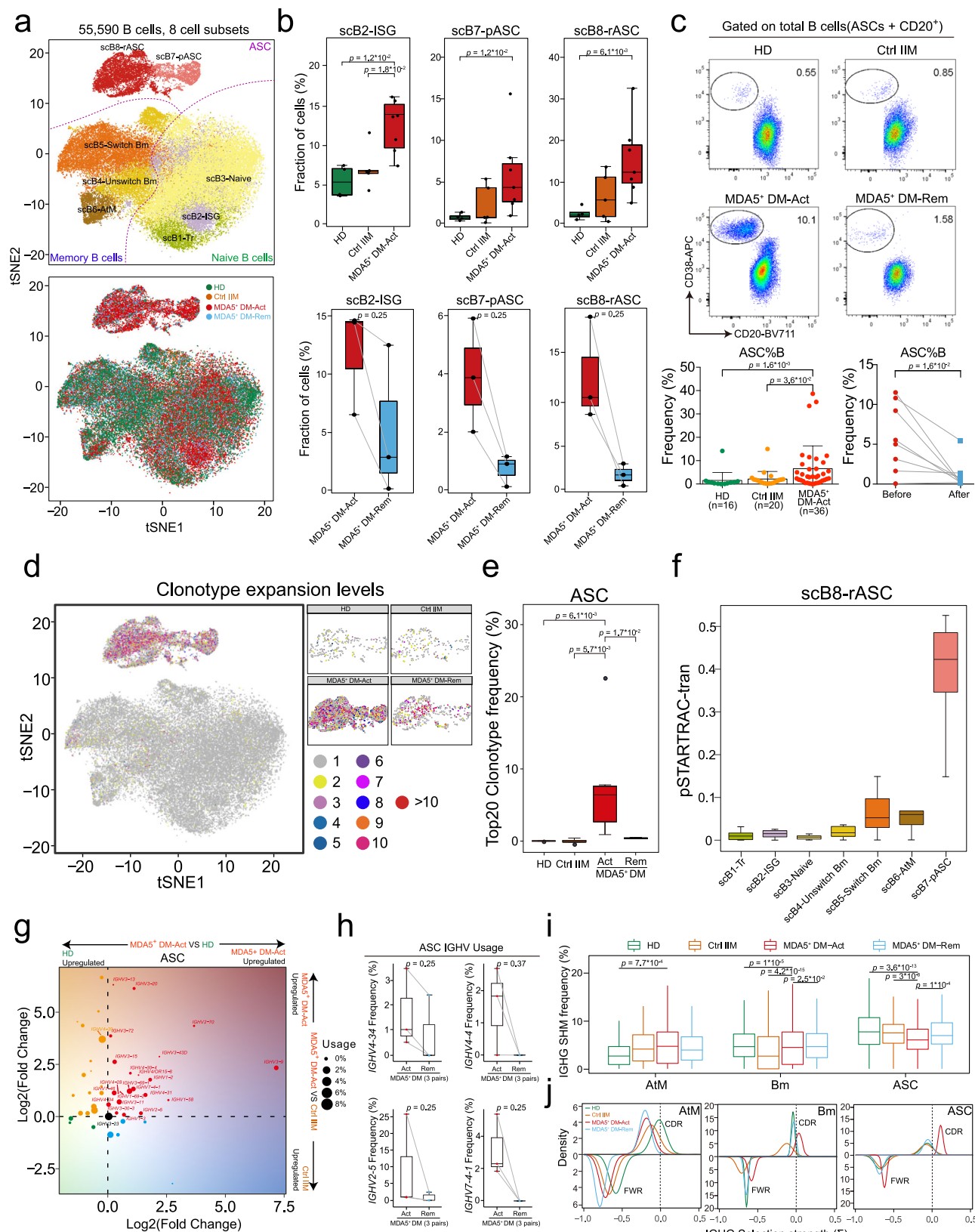

Behçet's disease[22] and pemphigus vulgaris[23]. *IGHV3-9* and *IGHV4-34* are overrepresented in systemic sclerosis[24] and systemic lupus erythematosus[22], respectively. Notably, the usages of *IGHV2-5, IGHV4-34, IGHV4-4*, and *IGHV7-4-1* were strongly decreased in the MDA5+ DM-Rem patients compared to the MDA5+ DM-Act patients (Fig. 2h and Supplementary Fig. 2d).

The detailed *IGH* features were further investigated in ASCs and Bm cells in the four groups. First, no particular differences in the frequencies of *IGH* isotypes (*IGHM, IGHA, IGHG*) (Supplementary Fig. 2e) or in the lengths of immunoglobulin heavy-chain complementarity-determining region 3 (Supplementary Fig. 2f) were observed in ASCs and Bm cells across the four groups. Second, when the somatic

**Fig. 2 | Expanded peripheral antibody-secreting cells (ASCs) with unique BCR features in MDA5⁺ DM patients. a** t-SNE plots showing eight B cell clusters, color-coded according to cell subset (upper) or group (lower). **b** Boxplots showing the proportions of scB2-ISG, scB7-pASC, and scB8-rASC cells from HD (*n* = 4), Ctrl IIM (*n* = 5), MDA5⁺ DM-Act (*n* = 7) and MDA5⁺ DM-Rem (*n* = 3) groups. **c** Representative flow cytometric plots showing the frequency of ASCs among all B cells from the indicated groups (upper). Scatter plots showing the accumulated data for the frequency of ASC from the indicated groups (bottom). Data are presented as mean ± standard deviation (SD) and the error bars denote SD (bottom left panel). **d** t-SNE plots showing the projection of BCR clonotype expansion levels on total B cells, color-coded by clonal expansion levels. **e** Boxplot showing the top 20 clonotype frequencies for each group. **f** Boxplot showing state transition between scB8-rASC and other B cell clusters in MDA5⁺ DM-Act patients quantified by the pSTARTRAC-tran algorithm according to the BCR clonotypes. **g** Inferred *IGHV* family usage preference in the ASCs of MDA5⁺ DM-Act patients compared with those from the HD (x-axis) or Ctrl IIM (y-axis) groups. Log2 (fold change) values were used to draw the plot, with each point sized by the *IGHV* usage and color-coded by the group specificity. **h** Box plots showing the comparisons of the frequencies of *IGHV4-34*, *IGHV4-4*, *IGHV2-5*, and *IGHV7-4-1* in ASCs between active and remitted MDA5⁺ DM patients (*n* = 3). **i** Boxplots showing the somatic hypermutation (SHM) frequencies of the *IGHG* genes of the Atm (*n* = 425), Bm (*n* = 2512) and ASC (*n* = 2329) cells across four groups. **j** Density plots showing the *IGHG* selection strengths on the complementary-determining region (CDR, upper) and framework region (FWR) of the indicated B cell clusters across four groups. Box plot center, box and whiskers correspond to median, interquartile range (IQR) and 1.5 × IQR, respectively (**b**, **e**, **f**, **h**, **i**). Statistical significance is calculated by the two-tailed Mann–Whitney test (**b**, **c**, **e**, **i**) and Wilcoxon matched-pairs signed rank test (**c**). Source data are provided as a Source Data file.

hypermutation (SHM) frequency of the antibody variable region was evaluated, a consistent finding was that the MDA5⁺ DM-Act group exhibited the lowest SHM frequencies among the four groups for both *IGHG* and *IGHA* in ASCs (Fig. 2i and Supplementary Fig. 2g). Interestingly, only the MDA5⁺ DM-Act group showed a clear positive selection for replacement mutations in complementary-determining region (CDR) regions in *IGHG* from the Bm cells and ASCs (selection strength > 0 means positive selection[25]) (Fig. 2j). An increased positive selection for replacement mutations was similarly observed in the CDR regions of *IGHA* in ASCs from the MDA5⁺ DM-Act group (Supplementary Fig. 2h). The SHM frequency and antigen-driven selection pattern are reminiscent of the isotype-switched Bm cells and ASCs observed in lupus patients[26], which suggests that autoantigen-driven antibody responses are strongly induced in MDA5⁺ DM-Act patients.

## Clonally-expanded CD8⁺ memory T cells in MDA5⁺ DM patients

T cells have been a therapeutic target in MDA5⁺ DM[4,19], and we analyzed the CD4⁺ and CD8⁺ T cell compartments in details. Nine CD4⁺ T cell clusters were defined by scRNA-seq (Supplementary Fig. 3a), and the MDA5⁺ DM-Act group was characterized by an increase in scCD4T2-ISG (Supplementary Fig. 3b, c). For CD8⁺ T cells, seven clusters were defined, and the proportions of three clusters (scCD8T4-GZMK⁺GZMB⁺ memory T cells (Tm), scCD8T6-ISG, scCD8T7-pTm [proliferating Tm]) were uniquely elevated in the MDA5⁺ DM-Act group compared with other groups (Fig. 3a, b and Supplementary Fig. 3d). We validated the increase of the scCD8T7-pTm cluster by detecting Ki67-expressing CD8⁺ T cells from an independent cohort (Supplementary Table 4). The MDA5⁺ DM-Act group (*n* = 32) had a significantly higher frequency of Ki67⁺CD8⁺ T cells compared with the HD (*n* = 16) and Ctrl IIM (*n* = 16) groups (Fig. 3c and Supplementary Fig. 3e). The frequencies of Ki67⁺CD8⁺ T cells were lower in the nine paired MDA5⁺ DM patients who have achieved remission (Fig. 3c).

TCR features were analyzed by scTCR-seq. CD4⁺ T cells exhibited limited clonal expansions (Fig. 3d and Supplementary Fig. 3f). In contrast, several CD8⁺ T cell clusters were clonally expanded (Fig. 3d and Supplementary Fig. 3g). In particular, the scCD8T4-GZMK⁺GZMB⁺ Tm, scCD8T6-ISG, and scCD8T7-pTm clusters from the MDA5⁺ DM-Act group showed the highest TCR clonalities among the four groups (Fig. 3e). Interestingly, the latter two clusters also showed the highest TCR diversities (Fig. 3e). TCR clonal overlap analysis revealed that there was high clonal sharing from scCD8T3 to scCD8T7 in the MDA5⁺ DM-Act group (Fig. 3f), suggesting that these Tm cell subsets were developmentally connected.

Given the highly expanded TCR clonotypes of CD8⁺ Tm cells observed in the MDA5⁺ DM-Act group, we investigated the fold changes in T cell receptor beta variable (*TRBV*) genes of CD8⁺ memory T cells between the MDA5⁺ DM-Act group and other groups. A panel of *TRBV* genes was strongly enriched in the MDA5⁺ DM-Act group (Fig. 3g), some of which were enriched in patients with systemic lupus erythematosus (*TRBV5-1, TRBV11-2, TRBV11-3*) and in those with

rheumatoid arthritis (*TRBV2, TRBV7-2, TRBV12-3, TRBV15, TRBV24-1, TRBV27*), or both diseases (*TRBV7-8, TRBV7-9*)[27]. Furthermore, we observed lower usages of *TRBV7-9, TRBV20-1, TRBV28, and TRBV30* in the MDA5⁺ DM-Rem patients compared to the MDA5⁺ DM-Act patients (Fig. 3h). These results indicate that CD8⁺ T cells are strongly activated and clonally expanded in active MDA5⁺ DM patients, and this response is reversed in remitted patients.

## Overactivation of the type I IFN signaling pathway in peripheral B and T cells in MDA5⁺ DM patients

To gain more insight into the molecular features of the four groups, a heatmap was generated by clustering the top 15 differentially expressed genes (DEGs) in peripheral B and T cells (Fig. 4a). No major difference in gene expression patterns was observed between B and T cells. Next, the upregulated DEGs from each group were used to perform gene ontology (GO) enrichment analysis. Notably, the MDA5⁺ DM-Act group was characterized by high expression of ISGs, namely *ISG15, MX1*, and *IFI6* (Fig. 4a and Supplementary Fig. 4a, b) and consequently, the GO terms were highly enriched with type I IFN signaling and related pathways (Fig. 4b). We also observed a positive correlation between the ISG scores and the clone frequencies of CD8⁺ T cell clusters in the MDA5⁺ DM-Act group ($R^2$ = 0.831) (Supplementary Fig. 4c, d), which is consistent with a report stating that type I IFNs promote T cell activation and proliferation[28].

For the Ctrl IIM group, we observed a panel of upregulated inflammation and stress-related genes, namely *FOS, FOSB, DUSP2*, and *GADD45B* (Fig. 4a and Supplementary Fig. 4a, b), and GO analysis revealed an enrichment of the p38 MAPK cascade and related inflammatory pathways (Fig. 4b). The gene expression pattern of the MDA5⁺ DM-Rem group was similar to that in the HD group, with enriched pathways related to homeostasis, including membrane protein proteolysis (Fig. 4b).

Transcription factors (TFs) orchestrate cell differentiation and function, and we analyzed TF regulatory networks (regulons) in peripheral B and T cells using single-cell regulatory network inference and clustering (SCENIC)[29]. As shown by SCENIC regulon areas under the curve per cell (AUCell) scores, the type I IFN-related TFs *IRF2/3/7/9* and *STAT1/2* were highly enriched in the MDA5⁺ DM-Act group, while the inflammation-related TFs, *FOS, FOSB, JUN*, and *KLF6* were enriched in the Ctrl IIM group (Fig. 4c, d). We modeled the regulatory network of these TFs to scan the DEGs for over-represented TF binding sites (Fig. 4e). In particular, we identified that *IRF3* and *IRF7*, which were overrepresented in the MDA5⁺ DM-Act group, constituted the hub regulons to control a large panel of downstream ISG genes (*MX1, ISG15, IFI6, OAS2*) (Fig. 4e). In contrast, *FOSB* and *KLF6*, which were overrepresented in the Ctrl IIM group, controlled the downstream genes involved in inflammation and stress (*NFKB1, NR4A3, CD44*, and *GADD45B*) (Fig. 4e).

We also measured the cytokines in plasma from the different groups. We found that plasma IFN-α concentration was uniquely

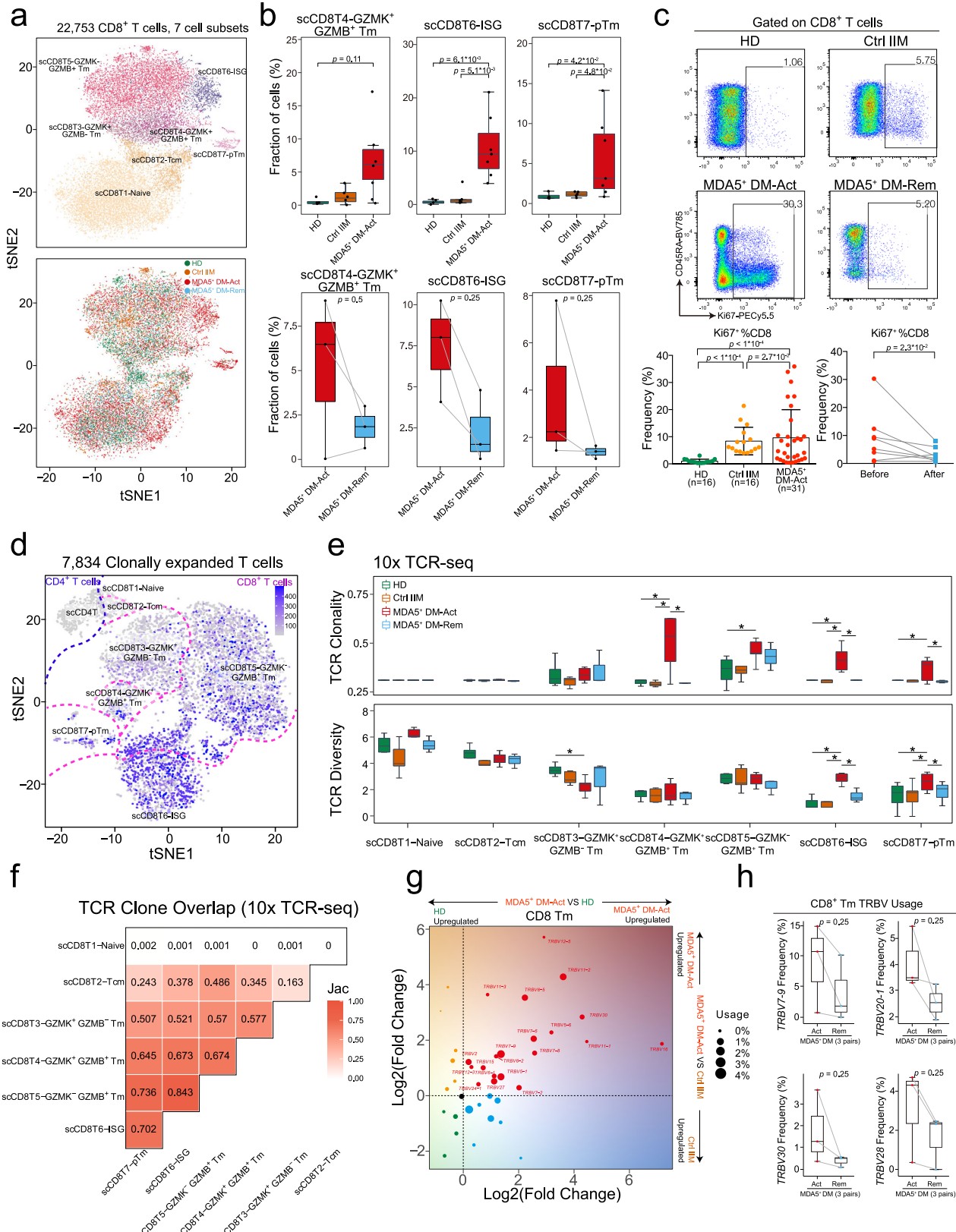

elevated in the MDA5+ DM-Act group compared with the Ctrl IIM and HD groups (Fig. 4f, g), confirming systemic activation of the type I IFN signaling pathway in the MDA5+ DM-Act patients. Inflammatory cytokines, namely IFN-γ, TNF-α, and IL-8 concentrations, were higher in both the MDA5+ DM-Act and Ctrl IIM groups than those in the HD group (Fig. 4f, g). One exception was vascular endothelial growth

factor D (VEGF-D) concentration, which was significantly higher in the Ctrl IIM group compared with other groups (Fig. 4f, g). Overall, by analyzing the transcriptome and TF network from peripheral B and T cells and plasma cytokines, the results further support that the type I IFN signaling pathway is highly activated in MDA5+ DM patients.

**Fig. 3 | Clonally-expanded CD8⁺ memory T cells and related TCR features in MDA5⁺ DM patients. a** t-SNE plots showing seven CD8⁺ T cell clusters, color-coded according to the cell subset (upper) or group (lower). **b** Box plots showing the proportions of scCD8T4-*GZMK*⁺*GZMB*⁺ Tm, scCD8T8-ISG, and scCD8T7-pTm cells from HD (*n* = 4), Ctrl IIM (*n* = 5), MDA5⁺ DM-Act (*n* = 7) and MDA5⁺ DM-Rem (*n* = 3) groups. **c** Representative flow cytometric plots showing the frequency of Ki67⁺ CD8⁺ T cells from indicated groups (upper). Scatter plots showing the accumulated data for Ki67⁺%CD8⁺ T cells from indicated groups (bottom). Data are presented as mean ± SD and the error bars denote SD (bottom left panel). **d** t-SNE plot showing 446 CD4⁺ T cells and 7397 CD8⁺ T cells with clonal expansion (clonotype >1), color-coded by the clonal expansion levels.**e** Box plots showing TCR clonality (upper) and TCR diversity (lower) in different CD8⁺ T cell clusters (*n* = 5666 cells; 1704 cells; 1285 cells; 542 cells; 4569 cells; 1175 cells; 519 cells; from left to right) across four groups. **f** Triangle heatmap showing the overlap of expanded TCR clonotypes across all

CD8⁺ T cell clusters in MDA5⁺ DM-Act patients. Data are aggregated from each patient within the group. Numbers indicate the normalized Jaccard index of shared expanded TCR clonotypes for each cluster pair. **g** Inferred *TRBV* family usage preference in the CD8⁺ Tm cells of MDA5⁺ DM-Act patients compared with those from the HD (x-axis) or Ctrl IIM (y-axis) groups. Log2 (fold change) values are used to draw the plot, with each point sized by the *TRBV* usage in MDA5⁺ DM-Act patients and color-coded by the group specificity. **h** Box plots showing the comparisons of the frequencies of *TRBV7-9*, *TRBV20-1*, *TRBV28*, and *TRBV30* in CD8⁺ Tm cells between active and remitted MDA5⁺ DM patients. Box plot center, box and whiskers correspond to median, IQR and 1.5 × IQR, respectively (**b**, **e**, **h**). Statistical significance is calculated by the two-tailed Mann–Whitney test (**b**, **c**, **e**) and Wilcoxon matched-pairs signed rank test (**c**). *, *p* < 0.05. Source data are provided as a Source Data file.

## Abnormal metabolic reprogramming of peripheral B and T cells in MDA5⁺ DM patients

Recently, we developed *scMetabolism*, a computational pipeline for quantifying single-cell metabolism[30], and we used this algorithm to map the metabolic pathways of peripheral B cells, and CD4⁺ and CD8⁺ T cells (Fig. 5a) and their subsets (Supplementary Fig. 5) across the four groups. We first ordered the top pathways according to the metabolic scores calculated from each immune cell in each group (Fig. 5a). Strikingly, we observed that B and T cells from the MDA5⁺ DM-Act group exhibited high metabolic activities compared with B and T cells from the three other groups (Fig. 5a). Several metabolic pathways, namely oxidative phosphorylation, pyruvate metabolism, glycolysis, pyrimidine metabolism and drug metabolism cytochrome P450, were heavily enriched in the MDA5⁺ DM-Act group (Fig. 5a, b) compared with other groups, and core genes from these pathways were also displayed (Fig. 5c).

Type I IFN signaling is involved in metabolic reprogramming in viral infections[31]. We performed a correlation analysis between ISG scores and the metabolic scores of peripheral B and T cells across the four groups. A positive correlation was observed between the two scores ($R^2$ = 0.73) (Fig. 5d), suggesting a role of type I IFNs in promoting metabolic reprograming in adaptive immune cells. Further ordering of the individual metabolic pathways according to Spearman's correlation coefficients revealed that oxidative phosphorylation and the citrate cycle were the top two pathways (Fig. 5e). Oxidative phosphorylation and the citrate cycle are involved in mitochondrial metabolism, which aids in adaptation to extracellular and intracellular stresses, such as nutrient and oxygen deprivation, and endoplasmic reticulum and oxidative stress[32]. Thus, the high metabolic activities observed in the MDA5⁺ DM-Act group's cells reflect a stressful status associated with abnormally elevated type I IFN signaling.

## Overactivation of type I IFN and fibrosis signaling pathways in the affected lungs of MDA5⁺ DM patients

To evaluate the microenvironment in the affected lungs of MDA5⁺ DM patients at single-cell resolution, we performed scRNA-seq on lung tissue from one MDA5⁺ DM patient who was undergoing lung transplantation (Fig. 6a). A total of 4337 cells from this MDA5⁺ DM patient and 9358 cells from the lungs of two healthy donors from a public source[33] were integrated and eighteen cell clusters were defined (Fig. 6b, Supplementary Fig. 6a and Supplementary Table 5). Even with only one MDA5⁺ DM patient sample, we observed noticeably elevated proportions of alveolar type II cells, fibroblasts, and endothelial cells within the non-immune cell compartment. For the immune cells, we saw increased proportions of ISG⁺CD4⁺ T cells, and ISG⁺CD8⁺ T and CD8 proliferating T cells (Fig. 6c). We then used mIHC to evaluate the relevant immune cells in lung biopsies from additional MDA5⁺ DM patients. Compared with the control lung tissues, high infiltration of immune cells and tertiary lymphoid structures were found in the lung tissues from MDA5⁺ DM patients (Fig. 6d). Furthermore, high

frequencies of Ki67⁺CD8⁺ T cells and MX1⁺CD8⁺ T cells were detected in the lungs of MDA5⁺ DM patients, irrespective of their infiltration degrees (Fig. 6d). These results indicate that potent adaptive immune responses are generated in the lungs of MDA5⁺ DM patients, accompanied with the strong activation of type I IFN signaling.

To further explore the potential pathogenic mechanisms in the affected lungs, single-sample gene set variation analysis (ssGSVA) was used to determine the specific pathways enriched in the MDA5⁺ DM patient. Several pathways, namely the type I IFN signaling pathway and fibrosis, were highly enriched in the lungs of the MDA5⁺ DM patient (Fig. 6e). Representative genes from both pathways were also displayed (Supplementary Fig. 6b, c). We further projected the ISG scores and fibrosis scores on the lung single-cell t-SNE maps from the MDA5⁺ DM patient and HDs. Generally, the lung cells from the HDs exhibited very low ISG and fibrosis scores. In contrast, high ISG scores were broadly distributed across different cell clusters in the MDA5⁺ DM patient. In addition, the fibrosis scores were mainly restricted to non-immune cells, particularly to the fibroblasts (Supplementary Fig. 6d). A tendency toward a positive association between the ISG and fibrosis scores in non-immune cells was also observed (Supplementary Fig. 6e).

Given the predominant type I IFN signaling signature within the lungs of the MDA5⁺ DM patient, we performed CellChat analysis[34] to further evaluate the type I IFN signaling network-based intercellular communications across different lung cell types. Interestingly, we saw that fibroblasts had strong type I IFN signaling pathway interactions with ASCs, CD8⁺ T cell subsets, and macrophages (Fig. 6f). We also performed CellPhoneDB analysis[35] to determine the ligand–receptor interactions between fibroblasts and several immune cell types (ASCs, ISG⁺CD4⁺T, ISG⁺CD8⁺T, and CD8 proliferating T cells). Extensive interactions were observed in the lungs of the MDA5⁺ DM patient, but not in HDs (Fig. 6g). Of note, several pairs, including TGFB1-TGF beta receptors1/2, were critically involved in fibrosis[36]. These data suggest that, in the context of overactivation of type I IFN signaling, the infiltrated immune cells and fibroblasts potentially form a unique profibrotic microenvironment in the lungs of MDA5⁺ DM patients.

## ISG15⁺CD8⁺ T cells as a promising prognostic biomarker for MDA5⁺ DM patients

The systemic activation of the type I IFN signaling pathway in MDA5⁺ DM patients prompted us to look for an accurate and convenient method to monitor this response. To this end, we have established a reliable flow cytometry method to measure ISG15⁺ and MX1⁺ B cells, CD4⁺ and CD8⁺ T cells after screening a panel of ISG molecules (Fig. 7a and Supplementary Fig. 7a, b). Compared with the Ctrl IIM and HD groups, the frequencies of ISG15⁺ and MX1⁺ B cells, CD4⁺ and CD8⁺ T cells were noticeably higher in the MDA5⁺ DM-Act group (Fig. 7b),and these frequencies were decreased in the MDA5⁺ DM-Rem group (Supplementary Fig. 7c). To evaluate whether these parameters can act as prognostic biomarkers, we performed a Cox hazard analysis for one-

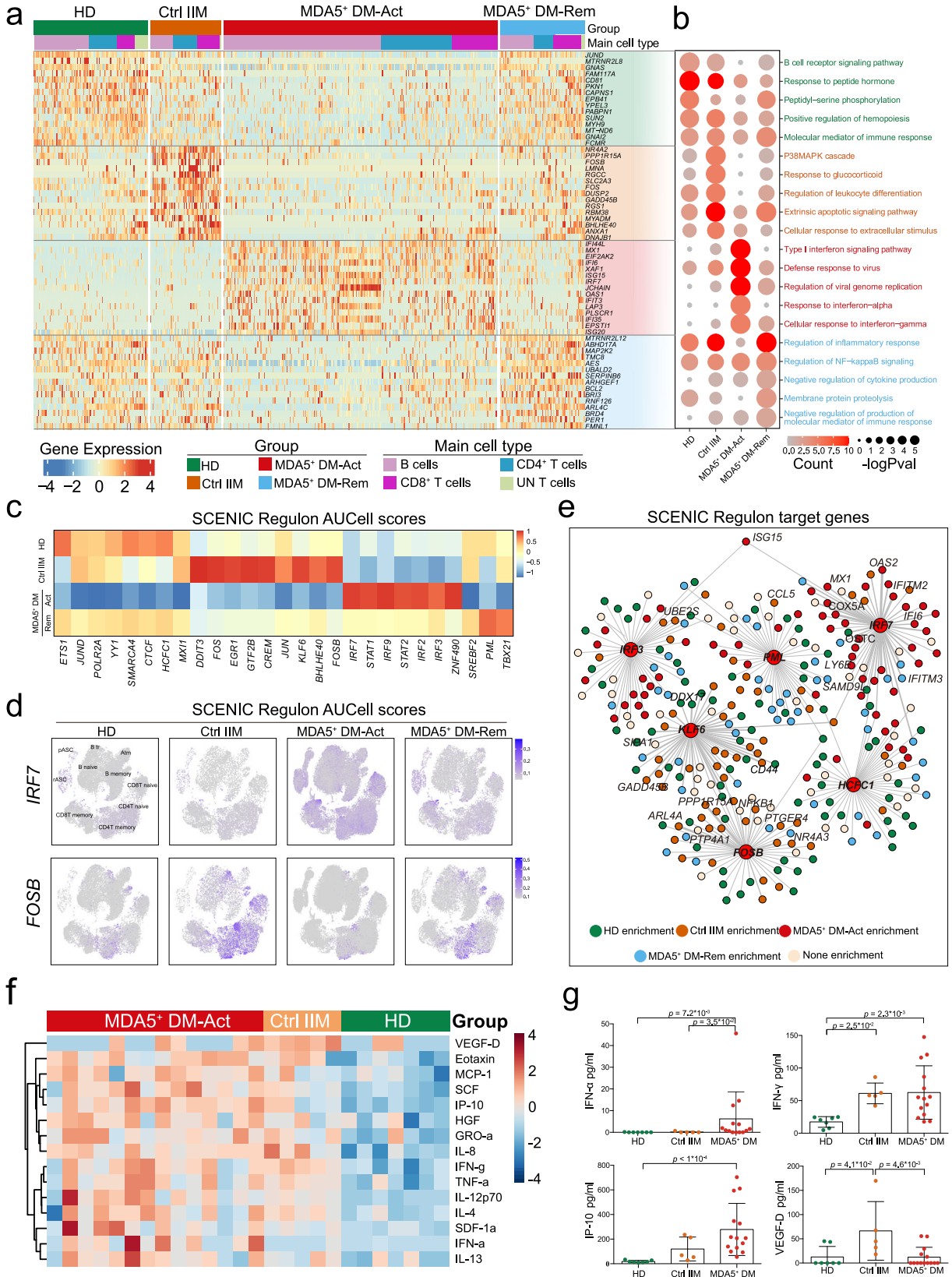

year survival in 31 MDA5+ DM-Act patients (Supplementary Table 4). Three routine laboratory parameters—serum lactate dehydrogenase (LDH), ferritin, anti-MDA5 titers were also included in the analysis. Univariate analysis showed that ISG15+%CD8+ T and MX1+%CD8+ T cells, and ferritin were significantly associated with one-year survival (Fig. 7c). Multivariate analysis further revealed that ISG15+%CD8+ T and

ferritin were independent prognostic markers (Fig. 7c). It should be mentioned that the anti-MDA5 titers from the MDA5+ DM patients at baseline did not show a prognostic value in this study, which is consistent with the other two studies that the baseline anti-MDA5 titers did not differentiate the patients who survived from those who succumbed to the disease[37,38].

**Fig. 4 | Transcriptional and regulon network analysis of peripheral B and T cells in MDA5⁺ DM patients. a** Heatmap displaying scaled expression values of the top 15 differentially expressed genes (DEGs) in peripheral B and T cells between the four groups. **b** Bubble plot showing the top 5 gene ontology (GO) term enrichment pathways from the upregulated top 15 DEGs of each group. **c** Heatmap showing the regulon areas under the curve per Cell (AUCell) scores from peripheral B and T cells across the four groups, calculated using the SCENIC algorithm. **d** t-SNE plots showing the regulon activities of *IRF7* and *FOSB* in peripheral B and T cells across the four groups, color-coded by AUCell scores. **e** Cytoscape graph showing the regulatory networks comprising regulons and their target genes underlying B and T cell development and function. For visualization, there are only 50 randomly-selected targets for each regulon. Each target gene is color-coded by the group specificity (genes with avglog2FC > 0.5 are marked in the graph). **f** Heatmap showing plasma cytokine profiles across the MDA5⁺ DM, Ctrl IIM, and HD groups. The scale bar represents the scaled average expression of the cytokines. **g** Scatter plots showing the plasma cytokine levels of IFN-α, IFN-γ, IP-10, and VEGF-D in the MDA5⁺ DM, Ctrl IIM, and HD groups. Data are presented as mean ± SD and the error bars denote SD. Statistical significance is calculated using the two-tailed Mann−Whitney test. avglog2FC: log of average expression fold change; IFN-α, interferon alpha; IFN-γ, interferon gamma; IP-10, inducible protein 10; VEGF-D, vascular endothelial growth factor-D. Source data are provided as a Source Data file.

Furthermore, compared with ferritin and other parameters, ISG15⁺%CD8⁺ T showed the highest area under the curve (AUC) value in the receiver operating characteristic (ROC) analysis to predict one-year survival (Fig. 7d). Using the optimal cut-off values set by the ROC analysis and the Kaplan−Meier survival analysis, we saw that the MDA5⁺ DM-Act patients with ISG15⁺%CD8⁺ T > 21.84% at baseline had significantly poorer one-year survival than those with ISG15⁺%CD8⁺ T < 21.84% (log-rank test, $P = 0.044$; Fig. 7e). Serum ferritin failed to show a significant difference between high-concentration and low-concentration groups by this survival analysis (log-rank test, $P = 0.11$; Fig. 7e). We also found that ISG15⁺%CD8⁺ T was highly correlated with other ISG parameters, but not with ferritin and LDH (Supplementary Fig. 7d), suggesting different natures of the two types of parameters. These analyses indicate that ISG15⁺%CD8⁺ T is a promising prognostic biomarker in comparison with the currently known parameters in MDA5⁺ DM.

We also evaluated the therapeutic efficacies in a retrospective cohort of active MDA5⁺ DM patients who received a single CNI ($n = 22$) or a combination of a CNI and tofacitinib ($n = 14$) (Supplementary Table 6). The 6-month survival percentage indicated that the combination treatment was associated with a significantly higher survival percentage than the CNI monotherapy (Fig. 7f). CNIs mainly target T cells, while tofacitinib, an oral inhibitor of Janus kinase 1 and 3, inhibits both the type I and II IFN signaling pathways[9]. Therefore, the superior efficacy in the combination treatment group indicates that more therapeutic benefits could be achieved in the active MDA5⁺ DM patients by targeting multiple cellular and molecular pathways.

## Discussion

MDA5⁺ DM represents an unmet medical need owing to the high incidence of RP-ILD and mortality, with little understanding of its aetiology and pathogenesis. Early cases of MDA5⁺ DM were reported in Japan[11,39,40] and China[41]; however, recently, substantial attention has been paid to this disease in Western countries[5,14,42,43]. In the current study, using single-cell and immune profiling, we identified that active MDA5⁺ DM patients display distinct cellular and molecular features in both B and T cell compartments. Furthermore, overactivation of the type I IFN signaling pathway is associated with aberrant metabolic remodeling in peripheral B and T cells and the profibrotic response in the affected lungs of MDA5⁺ DM-Act patients, providing novel insights into this complex disease.

Anti-MDA5 autoantibody is a key feature of MDA5⁺ DM that highlights the breach of B cell tolerance in this disease. However, except for a diagnostic role and an indicator of disease severity of the anti-MDA5 autoantibody[12,10,13], little is known about the humoral response in MDA5⁺ DM. With scRNA-seq and flow cytometry, we demonstrated that active MDA5⁺ DM patients were characterized by an increase of circulating ASCs. BCR analysis revealed that these ASCs were clonally expanded with autoimmune-prone features. Particularly interesting, isotype-switched *IGHG* and *IGHA* from ASCs showed antigen-driven selections. While the roles of anti-MDA5 and other potential autoantibodies require further investigation, an exacerbated humoral response is clearly demonstrated in MDA5⁺ DM patients. Several case reports have shown that B cell depletion therapy is effective in MDA5⁺ DM patients, especially in patients with RP-ILD who are refractory to conventional immunosuppressive agents[44−47]. Thus, our study provides direct evidence that the B cell arm should be considered in the clinical management of MDA5⁺ DM.

T cells have been a major therapeutic target, and CNIs are routinely included in the treatment of MDA5⁺ DM[4,19]. Our study revealed that several CD8⁺ T cell subsets, namely *GZMK⁺GZMB⁺*, ISG⁺, and proliferating CD8⁺ cells, are overrepresented in active MDA5⁺ DM patients. TCR repertoire analysis indicated that these CD8⁺ cell subsets were clonally expanded and developmentally connected. Furthermore, the expanded *TRBV* genes from CD8⁺ T cells also showed autoimmune-prone features in active MDA5⁺ DM patients. Of note, ISG⁺ and CD8⁺ pT cells were also highly enriched in the affected lungs of MDA5⁺ DM patients. These results highlight that CD8⁺ T cell response is preferentially activated, and CD8⁺ T cells could be a more precise target when developing future therapies.

The potent CD8⁺ T cell responses also provide clues regarding immune triggering in MDA5⁺ DM. The aetiology of MDA5⁺ DM remains elusive, and it is hypothesized that MDA5⁺ DM could result from certain viral exposures[48,49] in genetically susceptible individuals[50,51]. CD8⁺ T cells are activated by recognition of specific peptides presented by major histocompatibility complex (MHC) I molecules on antigen-presenting cells[52]. MHC I molecules present antigens to CD8⁺ T cells through the endogenous or cross-presentation pathways[53]. Interestingly, a truncated variant of *WDFY* family member 4 (*WDFY4*) with enhanced function is associated with MDA5⁺ DM[54]. *WDFY4* has been shown to be critical for antigen cross-presentation in XCR1⁺ classical dendritic cells[55], which raises an interesting possibility that truncated *WDFY4* may promote cross-presentation in MDA5⁺ DM. Future studies are needed to better understand the enhanced CD8⁺ T cell response in MDA5⁺ DM patients.

Activation of the type I IFN signaling pathway is a predominant feature in MDA5⁺ DM patients and is implicated in endothelial injury[56,57], vasculopathy[58,59], and lung injury[60,61]. Our study confirmed a strong molecular type I IFN signature in peripheral B and T cells and lung cells from active MDA5⁺ DM patients. In contrast, the Ctrl IIM patients exhibited preferential activation of the p38 MAPK pathway in peripheral B and T cells, revealing possibly different pathogenic natures between the two types of disease. Of note, type I IFNs have been shown to promote B cell terminal differentiation[62,63] and CD8⁺ T cell expansion[28]. Thus, we consider that type I IFNs may play direct roles in promoting the ASC response and CD8⁺ T cell proliferation in MDA5⁺ DM patients. We also provided evidence that type I IFNs may promote metabolic reprogramming of peripheral B and T cells and lung fibrosis in MDA5⁺ DM patients. We further showed that ISG15⁺ CD8⁺ T cells could be a promising prognostic biomarker in MDA5⁺ DM. While the triggering mechanism of type I IFN production is still under investigation, our study reinforces the idea that overactivation of the type I IFN signaling pathway is a key molecular feature in MDA5⁺ DM.

We also observed an increase of circulating IFN-γ in MDA5⁺ DM patients compared to HDs, which is consistent with several early reports[64,65]. The source of IFN-γ could be from multiple cell types, including T cells and NK cells. A recent study has indicated the role of

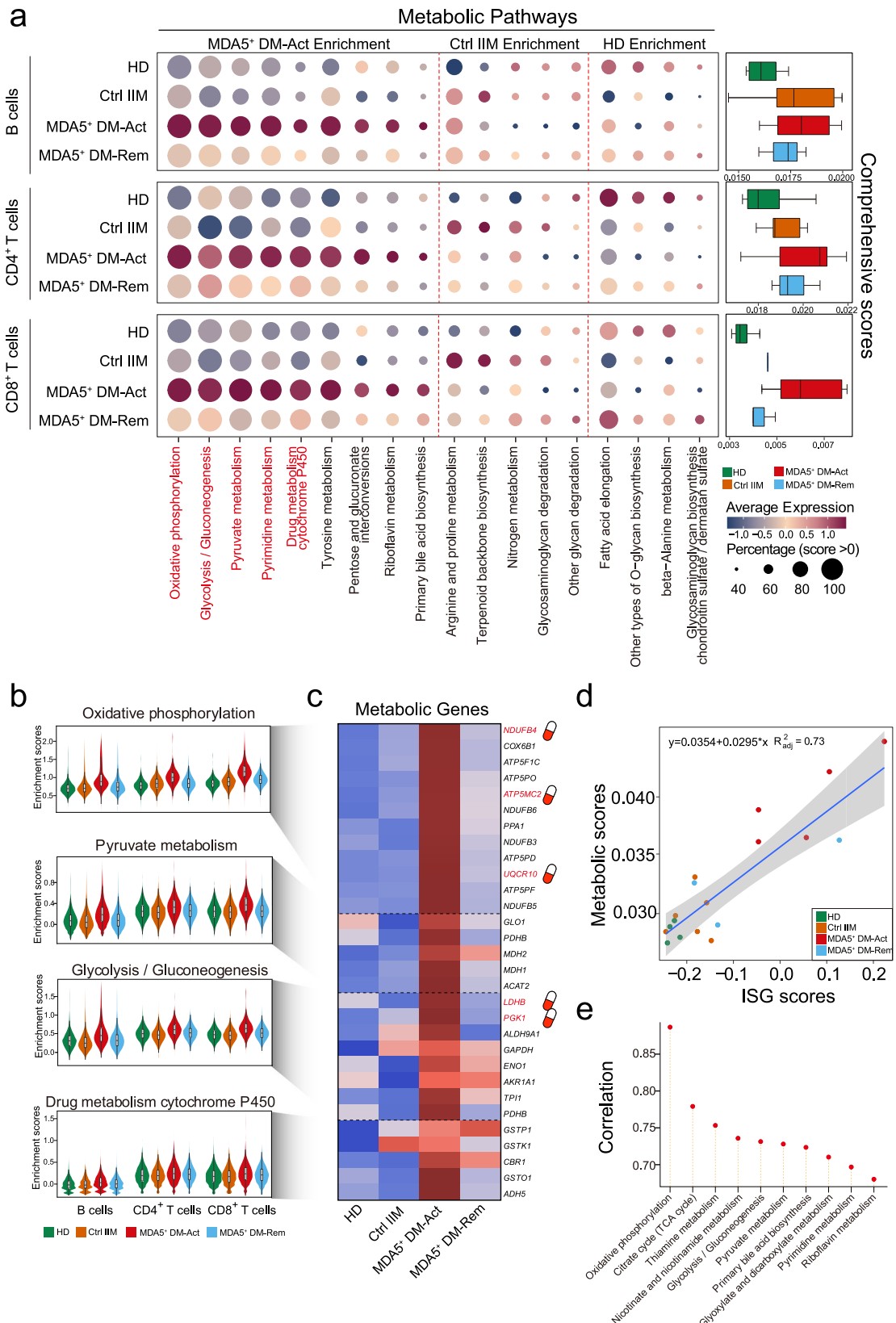

autoantibodies from MDA5⁺ DM patients to induce IFN-γ secretion in peripheral mononuclear cells, thus making an interesting link between autoantibody response and IFN-γ production[66]. A high level of IFN-γ is also implicated in the development and severity of MDA5⁺ DM. A positive correlation between serum IFN-γ levels and the computed tomography (CT) ground glass opacity (GGO) scores (G-scores)

indicates that IFN-γ may play an important role in the lung patho-physiology of MDA5⁺ DM[65]. Furthermore, IFN-γ and TNF-α synergisti-cally induce cytotoxicity in endothelial cells, which may contribute to the unique cutaneous lesions and potentially the pulmonary lesions observed in MDA5⁺ DM[67]. IFN-γ also synergizes with IL-1β to induce proinflammatory CX3CL1 in lung fibroblasts, which adds another

**Fig. 5 | Abnormal metabolic reprogramming of peripheral B and T cells in MDA5+ DM patients. a** Bubble plots showing the metabolic activities of peripheral B cells (n = 55,590), and CD4+ T cells (n = 30,242) and CD8+ T cells (n = 22,753) across the four groups. Color represents the scaled metabolic scores sized by the fraction of cells (ssGSVA score >0). The MDA5+ DM-Act group has the highest comprehensive metabolic scores (right). **b** Violin plots showing the selected metabolic pathways of B cells (n = 55,590), CD4+ T cells (n = 30,242), and CD8+ T cells (n = 22,753) across the groups. **c** Heatmap showing the average expression of selected genes from selected metabolic pathways as in (**b**). The potential druggable targets are labeled with a pill symbol. **d** Correlation analysis between metabolic scores and ISG scores by each donor from all four groups (Spearman's correlation) with a 95% confidence. **e** Ordering the individual metabolic pathways according to the Spearman correlation coefficients between each metabolic pathway score and ISG score. ssGSVA, single-sample gene set variation analysis; ISG, interferon-stimulating gene. Box plot center, box and whiskers correspond to median, IQR and 1.5 × IQR, respectively (**a**, **b**).

potentially pathogenic role of IFN-γ in MDA5+ DM[68]. As both type I and II IFNs activate Janus kinases, from a therapeutic point of view, the inclusion of Janus kinase inhibitors like tofacitinib[9] in the treatment regimen will have a combined advantage to inhibit both the type I and II IFN signaling pathways.

Given the similar clinical features between active MDA5+ DM and severe COVID-19[6], it is of particular interest to compare the cellular and molecular features between the two diseases. Notably, the severe COVID-19 patients also showed strong ASC and CD8+ T cell responses[69–71] similar to active MDA5+ DM patients. However, we are aware that there is clear difference between the two types of diseases. First, a predominant extrafollicular B cell response linked with ASC differentiation was observed in COVID-19[71,72], which we failed to see in active MDA5+ DM. Second, while we observed a very stong activation of type I IFN signaling pathway in peripheral B and T cells from active MDA5+ DM patients, this pathway was more marked in mild disease, and attenuated in severe disease from COVID-19 patients[70,71]. On the contrary, strong induction of the AP-1/p38MAPK pathway was consistently observed in severe COVID-19[71]. Clearly, more studies are needed to further explore the similarity and disparity between active MDA5+ DM and severe COVID-19.

Considering the complex nature of MDA5+ DM, we propose to define a patient's immune status before tailored therapy. To this end, we advocate real-time detection of circulating ISG15+ and MX1+ B and T cells and other relevant cell subsets by flow cytometry. These parameters could help guide the rational selection of therapeutic targets. For instance, Janus kinase inhibitors for the type I and II IFN signaling pathways, CNIs for the T cell arm, and rituximab for the B cell arm. This is of particular importance as overused immunosuppressive therapies may lead to unfavorable outcomes in MDA5+ DM patients[4].

The current study had several limitations. First, the number of samples used for single-cell sequencing was relatively small. In particular, the lung tissue was obtained from only one MDA5+ DM patient. Second, the current study mainly focused on the adaptive immune system. Third, there was only one cohort for the prognostic study. These limitations will be addressed in our future studies by establishing prospective study cohorts with more patients.

In summary, we described the landscape of the adaptive immune system in MDA5+ DM patients, and the identified cellular and molecular abnormalities might shed new light on the immunopathogenesis of MDA5+ DM. Real-time monitoring of patients' immune statuses and rational design of therapeutic targets might provide new opportunities to treat this insidious disease.

## Methods

### Ethical statement
This study was reviewed and approved by the Institutional Review Board of Renji Hospital (ID: 2013-126), Shanghai, China. Informed consent was obtained from all study participants. All studies were performed in accordance with the Declaration of Helsinki.

### Human specimens
Peripheral blood and lung tissue samples were collected from patients who were admitted in the Department of Rheumatology in Renji Hospital and fulfilled the 2017 EULAR/ACR Classification Criteria for Idiopathic Inflammatory Myopathies[1]. Patients' demographic information, as well as laboratory data, clinical manifestations, and treatments were recorded and presented in Supplementary Tables 1, 3, 4, and 6. Detailed information regarding patient myositis-specific antibodies (MSAs) was also presented.

Patients received immunosuppressive medications for their myopathy, and ongoing disease activity was measured by Myositis Intention-to-Treat Index (MITAX)[73]. MITAX, as described by the International Myositis Assessment & Clinical Studies Group, is a disease activity score with seven domains: cutaneous, muscle, constitutional, skeletal, gastrointestinal, pulmonary, and cardiovascular. The MITAX score was the sum points divided by the total 63 points. The details of clinical information required for the estimation of MITAX were obtained from the medical records. 9 points for a single domain generally indicate the intention to treat and indication for immunosuppression. We defined the score of 0.14 (9/63) or higher for any single domain as "MDA5+ DM Active (MDA5+ DM-Act)", and the score below 0.14 (9/63) as "MDA5+ DM Remission (MDA5+ DM-Rem)".

### Sample preparation
Human peripheral blood mononuclear cells (PBMCs) were isolated by Lymphoprep (Axis-Shield) density gradient centrifugation, and washed twice in MACS buffer (PBS supplemented with 5% fetal bovine serum (FBS, Gibco) and 2 mM EDTA (Gibco)). Plasma was isolated by centrifugation and the supernatant was collected prior to cryopreservation in −80 °C. The fresh lung tissue sample was cut into small pieces and then digested with 1 mg/ml collagenase IV and 0.1 mg/ml DNase I in RPMI-1640 medium for 1 h at 37 °C. Cells were filtered with 70 μm cell-strainer nylon mesh (Falcon) to remove undigested tissues and then centrifuged at 500 × g for 10 min. After centrifugation, the cell pellet was collected by gently removing the supernatant, and was washed twice with MACS buffer before staining.

### Flow cytometry
PBMCs were first stained with Zombie Yellow (Cat: 423104, Biolegend) to remove dead cells. For surface staining, cells were stained with fluorochrome-labelled antibodies in the staining buffer (PBS with 5% FBS, 2 mM EDTA, and 0.09% NaN3) at room temperature for 15 min. For intracellular staining, cells were fixed and permeabilized by a Cytofix/Cytoperm Fixation/Permeablization Kit (Cat: 554714, BD) and intracellularly stained with appropriated fluorochrome-labelled antibodies in the Permeablization buffer for 30 min at 4 °C. Cells were acquired on a flow cytometer (LSRFortessa, BD). Flow cytometry analysis was performed with Flowjo software (BD).

The following antibodies and reagents were used in the study: FITC anti-human CD3 (Cat: 300406, Clone: UCHT1, Biolegend), BUV395 anti-human CD3 (Cat: 564001, Clone: SK7, BD), PerCPCy5.5 anti-mouse CD3 (Cat: 100218, Clone: 17A2, Biolegend), APCCy7 anti-human CD4 (Cat: 357416, Clone: A161A1, Biolegend), Alexa Fluor 700 anti-human CD8 (Cat: 300920, Clone: HIT8a, Biolegend), BV570 anti-human CD14 (Cat: 301832, Clone: M5E2, Biolegend), PE anti-human CD19 (Cat: 302254, Clone: HIB19, Biolegend), BV711 anti-human CD19 (Cat: 302246, Clone: HIB19, Biolegend), BV785 anti-human CD19 (Cat: 302240, Clone: HIB19, Biolegend), BV711 anti-human CD20 (Cat: 302342, Clone: 2H7, Biolegend), PECy7 anti-human CD27 (Cat: 302838, Clone: O323, Biolegend), BV650 anti-human CD27 (Cat: 302828, Clone: O323, Biolegend), APC anti-human CD38 (Cat: 303510, Clone: HIT2,

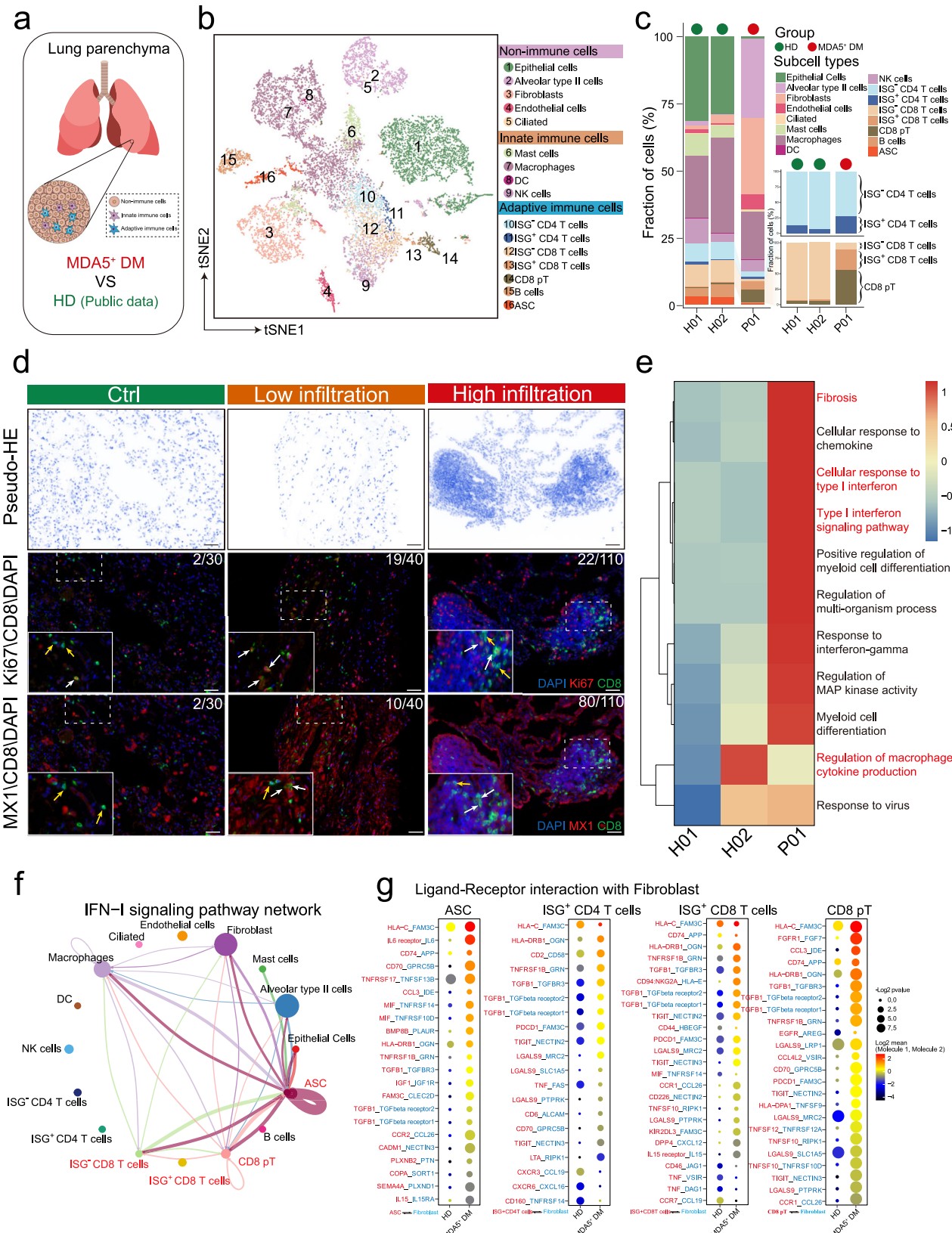

Biolegend), BV785 anti-human CD45RA (Cat: 304140, Clone: HI100, Biolegend), Biotin anti-human TCRgd (Cat: 331206, Clone: B1, Biolegend), Alexa Fluor 488 anti-human MX1 (Cat: ab237298, Clone: EPR19967, Abcam), PE anti-human ISG15 (Cat: IC8044P, Clone: 851701, R&D), PECy5.5 anti-human Ki67 (Cat: 35-5699-42, Clone: 20Raj1, eBioscience), Streptavidin BV570 (Cat: 405227, Biolegend). The

dilution for fluorophore-conjugated antibodies was routinely 1:100, and the dilution for Streptavidin BV570 was 1:500.

### Cytokine detection and heatmap presentation
Cytokines in the plasma were detected by the multiplexed Luminex xMAP assay (Cytokine/Chemokine/Growth Factor 45-Plex Human

**Fig. 6 | Single-cell profiling of the lung tissues from an MDA5+ DM patient.**
**a** Diagram depicting the lung sample sources. Adobe Illustrator 2020 was used to create the image. **b** t-SNE plot showing of the overview of 16 cell clusters in the integrated single-cell transcriptomes of 13,695 lung cells from one MDA5+ DM-Act patient (4337 cells) and two healthy donors (9358 cells) from a public source. Clusters are named as indicated cell subsets according to the specific gene expression patterns (Supplementary Table 3) and are color-coded. **c** Comparison of cell type fractions in individual samples. CD4+ and CD8+ T cell subsets are shown on the right. **d** Representative mIHC images showing the staining for CD8 (green, middle and bottom), Ki67 (red, middle), MX1 (red, bottom), and DAPI (blue, middle and bottom) in the lung tissues of a Ctrl (n = 1; normal part of lung adenocarcinoma)

and MDA5+ DM (n = 2; low immune cell infiltration and high immune cell infiltration, respectively) patients. The white arrows indicate Ki67+CD8+ T cells (middle) and MX1+CD8+ T (bottom) cells, and the yellow arrows indicate Ki67–CD8+ T cells (middle) and MX1–CD8+ T cells (bottom). Scale bar: 50 μm. **e** Heatmap showing GO enrichment pathways for individual donors according to donor-specific expressed genes (avglog2FC > 0.5). **f** CellChat analysis showing type I IFN signaling inter-cellular communication between cell subtypes. **g** Bubble plots showing ligand–receptor interactions between fibroblasts and the indicated cell types, with a significant difference between the MDA5+ DM and HD groups. mIHC, multiplex immunohistochemistry; DAPI, 4', 6-diamidino-2-phenylindole; Ctrl, control; GO, gene ontology; avglog2FC: log of average expression fold change.

Panel 1 (eBioscience) according to the manufacturer's instruction. The differentially expressed cytokines were first identified based on analysis of variance (ANOVA) using the R function aov, and the differences between groups were tested using Tukey's range test implemented in R function TukeyHSD (p-value adjusted by Tukey's 'Honest Significant Difference' method <0.05). Pheatmap (V1.0.12) package and ward.D2 hierarchical cluster analysis were used to illustrate the cytokine difference between different groups.

## Single-cell RNA sequencing
PBMCs were stained by anti CD3 and anti CD19 in the staining buffer. Cells were filtered using 40 μm cell-strainer nylon mesh (Falcon) and DAPI (Sigma) was added before sorting. CD3+CD19– T cells and CD3– CD19+ B cells were sorted on a FACSAria II (BD Biosciences) and the purity was routinely more than 98%. The cell suspension of each sample was subjected to the Chromium Next GEM Single Cell 5' Reagent Kit (V2) (10x Genomics, Pleasanton, CA) to prepare single cell 5' gene expression libraries, 5' cell surface protein libraries and V(D)J libraries following the manufacture's protocols (10× Genomics). For lung cell scRNA-seq, the Chromium Next GEM Single Cell 3' Reagent Kit (V2) (10x Genomics, Pleasanton, CA) was used. The single cell libraries were sequenced on Illumina NovaSeq 6000 Systems using paired-end sequencing (150nt).

## Single-cell transcriptome data processing
All of single-cell transcriptome sequencing data were aligned and quantified by using Cell Ranger (V6.1.1, Linux) against the GRCh38 human reference genome from 10x Genomics official website. The preliminary counts were then used for downstream data analysis by Seurat (V4.0.2, R), including quality control, normalization, feature selection, dimension reduction, unsupervised clustering, and visualization[74]. Quality control was performed to remove the low-quality cells, with less than 500 detected genes or more than 10% mitochondrial gene counts. Then, we performed DoubletFinder (V2.0.3, R) to identify and remove the potential doublets[75]. The standard of identifying the doublet cells for each single cell was based on the bimodal distribution by using the default parameters. We performed data normalization by employing a global-scaling normalization method "LogNormalize" that normalized the feature expression measurements for each cell by the total expression, multiplied this by a scale factor (10,000), and log-transformed the result. As for lung tissue data, we integrated our data with the public lung scRNA-seq data from two health donors[33] by the same pipeline.
We performed the function FindVariableFeatures (Seurat) to detect the features with highest coefficient of variation (CV). By default, we chose the top 2000 variable features to calculate a PCA matrix with 40 components and transported the PCA matrix into Harmony (V0.1.0, R) to integrate single-cell data gene expression matrix and correct the batch effect[76].

## Unsupervised clustering and dimensional reduction
The Harmony matrix would be used for unsupervised clustering by building the nearest neighbor graph and Louvain algorithm. The first round of unsupervised clustering (resolution = 0.6) for identifying the

main cell types, including the B cells (CD79A), ASC (XBP1), CD4+ T cells (CD4), CD8+ T (CD8A) cells, and unconventional T cells (TRDC). The second round of clustering in PBMC samples was performed in each main cell type (resolution = 1). As for the lung samples, we performed the same procedure with all cells (resolution = 1). Lung cells were classified into 16 clusters, including epithelial cells, alveolar type II cells, fibroblasts, endothelial cells, ciliated, mast cells, DC, NK cells, CD4 T cells, ISG+ CD4 T cells, CD8 T cells, ISG+ CD8 T cells, pCD8 T cells, B cells, and ASC.
To visualize the relationships in cell clades and the significant difference in four groups, we performed clustering analysis on PBMCs' scRNA-seq data with TooManyCells (V2.2.0.0, Linux)[20]. In detail, we set the minimum size of a leaf to 2000 cells in a case and colored each leaf by disease status. The background region was added to highlight the disease-specific regions. Group preference of cell types was estimated by STARTRAC-dist (v0.1.0) index.

## Differential gene expression and pathways enrichment analyses
To calculate the significant marker genes for each cell subtype and group, we performed the FindAllMarkers function (Seurat) to identify differentially expressed genes (DEGs) (|Log2FoldChange| > 0.5, P-value adjust <0.05).
To annotate the function of these DEGs, we performed the Gene Ontology (GO) terms pathway enrichment analysis by using the ClusterProfiler (V3.19.0, R) following the default parameters (datasets: org.Hs.eg.db V3.12.0)[77]. The HALLMARK and KEGG gene sets were from the molecular signatures database (MSigDB, V7.4).

## ISG score, fibrosis score, and comprehensive metabolism score
To evaluate the type-I interferon and fibrosis activities, we performed the AddModuleScore (Seurat) function based on the interferon-related module genes in Systemic Lupus Erythematosus (SLE)[78] and the reported fibrosis-related genes in lung diseases[79]. The reference genes to calculate ISG and fibrosis scores were listed in Supplementary Table 7. As for the comprehensive metabolism score, we followed the same pipeline based on the top20 activated metabolic pathways.

## SCENIC analysis
To predict the potential transcriptional regulatory network in B and T cells, we performed the SCENIC analysis[29], using the 10-thousand motifs database for RcisTarget (V1.10.0, R), GRNboost2 (V0.1.6, python), and AUCell (V0.99.5). The input matrix was the normalized expression matrix, output from Seurat. After calculation, the result matrix of transcription factor AUCell scores was integrated into the Seurat S4 object. Restricted by the layout, only 50 regulated targets of each transcription factor were sampled and visualized by Igraph (V 1.3.0, R).

## Single-cell metabolism analysis
To quantify single-cell metabolic activity, we performed the scMetabolism[80] (V0.2.1, R). For fast running and applying in large datasets, we chose the VISION pipeline to evaluate the KEGG metabolism gene sets signature scores. The resulting matrix of metabolism gene sets signature scores was integrated into the Seurat S4 object too.

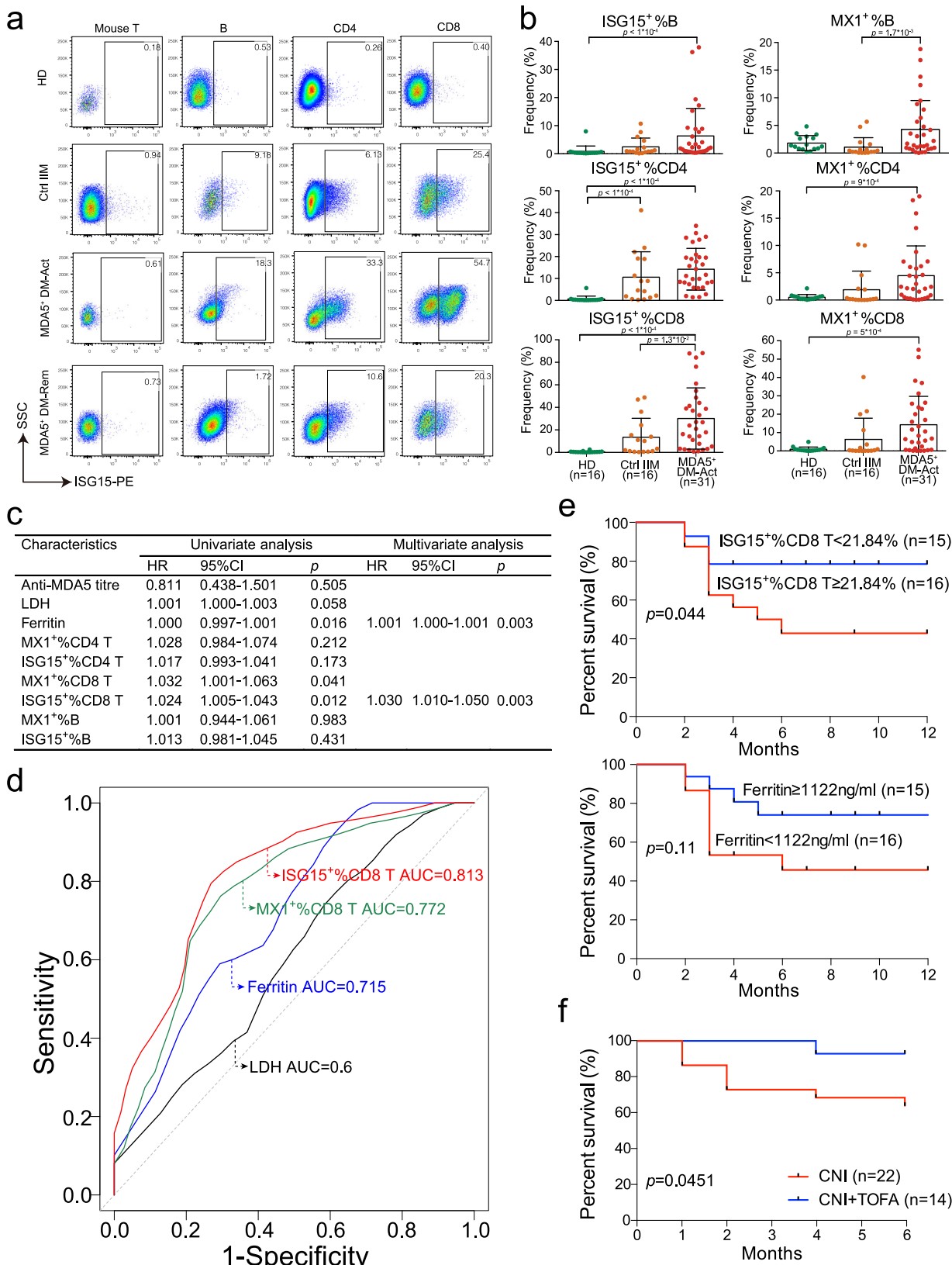

## Intercellular ligand-receptor analysis

To search for the interaction with lung fibrosis-related cell subtypes, we performed the CellChat (V1.1.0, R)[34] to evaluate the IFN-I signaling pathway network intercellular communication in lung microenvironments. Furthermore, we performed CellPhoneDB (V2.0, Linux)[35] to infer the ligand-receptor interaction between fibroblasts with other

cell types. The top 20 activated ligand-receptor pairs in the MDA5[+] DM-Act group were shown.

## BCR and TCR repertoire analysis

All BCR/TCR sequences were assembled and quantified following Cell Ranger vdj pipeline against GRCH38 reference genome. The low-

**Fig. 7 | Prognostic value of ISG15⁺CD8⁺ T cells. a** Representative flow cytometric plots showing the gating strategies for ISG15⁺ B cells, and CD4⁺ and CD8⁺ T cells. Mouse CD3⁺ T cells were used as the negative controls. **b** Scatter plots showing the frequencies of ISG15⁺%B, ISG15⁺%CD4, ISG15⁺%CD8, MX1⁺%B, MX1⁺%CD4, and MX1⁺%CD8 cells from the indicated groups. The number of each group is indicated in parentheses. Data are presented as mean ± SD and the error bars denote SD. Statistical significance is calculated using the two-tailed Mann–Whitney test. **c** Univariate and multivariate Cox proportional hazards analyses of the indicated parameters for 1-year survival in active MDA5⁺ DM patients (*n* = 31). **d** Receiver operating characteristic (ROC) analysis of the indicated parameters to predict 1-year survival in active MDA5⁺ DM patients (*n* = 31). **e** Kaplan–Meier analysis of 1-year survival for ISG15⁺%CD8 and ferritin in MDA5⁺ DM patients (*n* = 31). The log-rank test is used. **f** Kaplan–Meier analysis of 6-month survival to compare the therapeutic efficacies in MDA5⁺ DM patients between the CNI group (*n* = 22) and the CNI + TOFA group (*n* = 14). The log-rank test is used. CNI, calcineurin inhibitor; TOFA, tofacitinib; ISG, interferon-stimulating gene. Source data are provided as a Source Data file.

quality cells labeled as low-confidence, non-productive or with less than 2 UMIs were removed. Only the B cells with one heavy chain and one light chain were retained. Similarly, only the T cells with one TCR α-chain and one TCR β-chain were retained. For further analysis, we perform the change-O[81] to annotate the VDJ sequences output from the Cell Ranger. This pipeline used the standard IMGT reference database of human alleles. The TigGer (V1.0.0, Linux) was performed to infer novel alleles and subject-specific genotypes[82]. To group sequences into inferred clonal groups, we clustered BCR sequences that had the same heavy chain V and J genes and the same junction length. We clustered sequences with similar junction regions, using a defined sequence distance cutoff (Hamming distance <0.1). The SHazaM package was used to evaluate the somatic hypermutation (SHM) and selection strength[25,81]. Finally, the pSTARTRAC-tran algorithm (V0.1.0, R) was used to calculate lineage tracking by clonotypes.

### TCR/BCR clonality and diversity analysis
TCR/BCR diversity[83] was calculated as Shannon's entropy shown below:

$$H(x) = -\sum_{i=1}^{N} p(x_i) \log p(x_i) \qquad (1)$$

The $p(x_i)$ represents the frequency of a given TCR/BCR clone $i$ among all T/B cells with TCR/BCR identified.

Jaccard index[84] was calculated as $|A \cap B|/|A \cup B|$, where A is the CDR3aa number of A subcluster of T/B cells and B is the CDR3aa number of B subcluster of T/B cells;

$$\text{Clonality} = 1 - H(x)/\log_2(N) \qquad (2)$$

The shannon entropy $H(x)$ was normalized entropy over the number of unique clones and $N$ is the number of unique clones.

### Multiplex immunohistochemistry (mIHC)
Lung tissues used for mIHC were from hospital tissue bank, including the needle biopsies for diagnosis from later confirmed MDA5⁺ DM patients and adjacent normal tissues from lung adenocarcinoma resections. 4-μm-thick sections were prepared from tissue blocks, and sections were deparaffinized in xylene and rehydrated in ethanol. mIHC was performed according to manufacturer's instruction (Perkin-Elmer Opal Kit). After antigen retrieval was conducted in preheated EDTA (pH 9.0) solution at 95 °C for 10 min, samples were placed aside until they were restored to room temperature. And then endogenous peroxidase activity was blocked by 3% H₂O₂ for 20 min, followed by goat serum (Vectorlabs) blocking for another 20 min. The primary antibody (1:1000 dilution) was incubated for 1 h at room temperature. After washing with TBS-T (Tris Buffered Saline Buffer with 0.05% Tween-20) three times, sections were incubated with ImmPRESS HRP Goat Anti-Rabbit IgG Polymer (Cat: MP-7451-50, Vectorlabs) or Goat Anti-Mouse IgG Polymer (Cat: MP-7452-50, Vectorlabs) secondary detection antibody (no dilution) for 20 min and were visualized using OPAL dyes (PerkinElmer Inc.). The sections were performed as previously described for the next few antibodies, which was again heated in citric acid buffer (pH 6.0) and incubated with the goat serum,

primary antibodies, secondary antibodies and OPAL dyes. The panel was conducted as follows: anti-human CD8 (Cat: M7103, Clone: C8/144B, Dako)-OPAL 520; Ki-67 (Cat: ab16667, Clone: SP6, Abcam)-OPAL 690; MX1(Cat: 13750-1-AP, Clone: Rabbit polyclonal IgG, Proteintech)-OPAL 570. Following nucleus staining with DAPI (Sigma), slides were scanned and imaged using the Vectra3 platform (Akoya). Data were analyzed by the inForm Automated Image Analysis Software (Akoya).

### Statistical analysis
Data were expressed as the mean ± standard deviation (SD), if not otherwise indicated. Statistical analysis was performed by Prism 9.0 (GraphPad). Comparisons were analyzed using Mann–Whitney test, or Wilcoxon signed–ranked test as appropriate. The Youden index was used for setting optimal cutoff. Univariate and multivariate analyses of the Cox proportional-hazards model were performed to identify independent prognostic risk factors. Kaplan-Meier curves were estimated by log-rank test. All statistical tests were two-tailed and $p < 0.05$ was considered significant.

### Reporting summary
Further information on research design is available in the Nature Research Reporting Summary linked to this article.

### Data availability
The single cell sequencing datasets of human healthy lungs used in this paper are publicly obtained from the Human Cell Atlas Data Coordination Platform/NCBI BIOPROJECT with the accession code PRJEB31843. The raw single-cell RNA sequencing data reported in this paper have been deposited at the National Genomics Data Center under the accession number HRA003082. The raw single-cell immune repertoire sequencing data have been deposited at the National Center for Biotechnology Information Sequence Read Archive under the accession number PRJNA881644. The raw sequencing and single-cell immune repertoire data are available for non-commercial purposes. Source data are provided with this paper. The remaining data are available within the Article or Supplementary Information.

### Code availability
The VDJ related codes supporting the current study are publicly available on Github (https://github.com/Zechuan-Chen/scVDJplot).

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

## Acknowledgements

We thank Jane Charbonneau, DVM, from Liwen Bianji (Edanz) (www.liwenbianji.cn) for editing the English text of a draft of this manuscript. This study was supported by the National Key Research and Development Program of China (No. 2021YFE0200600 to X.M.Z.); the Strategic Priority Research Program (No. XDPB0303 to X.M.Z.), Chinese Academy

of Sciences; National Natural Science Foundation of China (Nos. 81771733 to C.D.B., 31770960 to X.M.Z. and 82201979 to Y.Y.); Shanghai Municipal Science and Technology Fund (Nos. 21ZR1438800 and 22Y11902400 to Q.F., 22490760100 to X.M.Z.) and Shanghai Talents Development Fund (No. 2019092 to Q.F.); Shanghai Municipal Science and Technology Major Project (Nos. 2019SHZDZX02 and HS2021SHZX001 to X.M.Z.), and the Shanghai Jiaotong University "Star of Jiaotong University" Medical Engineering Cross Research Fund (No. YG2022QN018 to Y.Y.) and and Pujiang Rheumatism Young Doctor Training Program (SPORG2108 to Y.Y.).

## Author contributions

Conceptualization, X.M.Z., Q.F., L.J.L., S.Y., and N.S.; Resources, Y.Y., X.L., X.Y.L., J.X. and C.D.B.; Methodology, Y.Y., Z.C.C., F.Y.J., T.L., P.H., and J.Q.M.; Investigation, Y.Y., S.J., X.Y.L., F.Y.J., J.X. and T.L.; Formal Analysis, Z.C.C., S.J., and F.Y.J.; Validation, S.J., F.Y.J., T.L., and J.Q.M.; Data Curation, Z.C.C., S.J., and P.H.; Visualization, Z.C.C. and P.H.; Writing-Original Draft, Y.Y., Z.C.C., and X.M.Z.; Writing-Review & Editing, X.M.Z., Q.F., and C.B.; Supervision, X.M.Z. and Q.F.

## Competing interests

The authors declare no competing interests.
