## [Peer Review File · Nature Communications]

Single-cell profiling reveals distinct adaptive immune hallmarks in MDA5⁺ dermatomyositis with therapeutic implicationsREVIEWER COMMENTS

Reviewer #1 (Remarks to the Author):

In this manuscript, Ye et al investigated by single-cell RNA sequencing the repertoire of circulating B and T cells of patients with MDA5+ DM. They also performed sc-RNA sequencing on one lung of a patient. By this way, they found that active MDA5+ DM patients display distinct cellular and molecular features in both B and T cell compartments. MDA5+ DM is a rare autoimmune disease with poor prognosis and thus any advance in our knowledge of the fundamental mechanisms involved in this disease is clinically important and valuable.

I have specific points to address:

- Results from scRNA-seq on B and T cells from active MDA5+ DM patients are compared to IIM disease controls (line 136 of the manuscript). However, it is not specified whether the patients in this control group are also in an active form of the disease. This can significantly change the interpretation of the results. A disease activity column (Active/remission) should appear in the Table S1.

- Line 301 : the authors state that plasma IFN- α concentration was uniquely elevated in the MDA5+ DM-Act group compared with the Ctrl IIM and HD groups. However, FIG. 4G clearly shows that one and only one plasma is very largely above the others in terms of concentration. I'm not sure that the difference between the MDA5 group and the IIM control group is statistically significant without this plasma.

- Line 304 : The results of the IFN-gamma quantifications are interesting but difficult to understand, because the results presented in Fig.1E suggest a decrease in Th1 responses in the MDA5 group compared to the control groups. However, several studies have reported a higher production of IFN-gamma in MDA5 patients, which seems to correlate with the activity of the disease. This point should be discussed in depth by the authors, as it may result from a methodological limitation. From a global point of view, the whole discussion is centered around IFN-I, but nothing is said about IFN-gamma.

- Line 308: Again, the vEGF concentration is statistically different in the IIM group, but that's surely only based on the value of only one plasma....

- Line 391 : the authors state that ISG15+ CD8+ T cells are a promising prognostic biomarker for MDA5+ DM patients, and they show higher frequencies of these cells in the DM-Act group. Once again, it is important to know the clinical activity of patients in the IIM group to know if this increased frequency of ISG15+ CD8+ T cells is specific to patients with MDA5+ DM, or if it simply reflects an active form, whatever the type of DM. In this part of the manuscript, the authors should also add anti-MDA5 antibody titers, since it is said that they have a predictive value (line 104 Introduction section).

- Line 423 : Tofacitinib also acts on the production of IFN-gamma, not only on the type I IFN signaling pathway. Since the authors have demonstrated high concentrations of IFN-gamma in the plasma of MDA5+ DM patients, this point is also to be discussed.

Reviewer #2 (Remarks to the Author):

The authors described in the present study the molecular features of MDA5+ DM patients by single-cell and BCR/TCR profiling.

They applied pretty standardized single-cell analysis methodologies; they were quite stringent in the criteria they used to select single cells for downstream analyses, which implies they have a high-quality dataset to work with. Reported statistical analysis is appropriate.

The limitation on the sample number, referred to in the discussion, is quite common for single cell studies, but the authors do not to make any overstatement about their findings, and are aware that further validations on other patient cohorts will be needed.

Overall, I find the manuscript well written, with a clear outline and well discussed.

Point-by-point Responses to the Reviewers' Comments

REVIEWER COMMENTS

Reviewer #1 (Remarks to the Author):

In this manuscript, Ye et al investigated by single-cell RNA sequencing the repertoire of circulating B and T cells of patients with MDA5+ DM. They also performed sc-RNA sequencing on one lung of a patient. By this way, they found that active MDA5+ DM patients display distinct cellular and molecular features in both B and T cell compartments. MDA5+ DM is a rare autoimmune disease with poor prognosis and thus any advance in our knowledge of the fundamental mechanisms involved in this disease is clinically important and valuable.

I have specific points to address:

- **Results from scRNA-seq on B and T cells from active MDA5+ DM patients are compared to IIM disease controls (line 136 of the manuscript). However, it is not specified whether the patients in this control group are also in an active form of the disease. This can significantly change the interpretation of the results. A disease activity column (Active/remission) should appear in the Table S1.**

Response:

We thank the reviewer's thoughtful advice. We have added one column of "Disease Activity/MITAX" in Table S1. MITAX denotes "Myositis Intention-to-Treat Index" which reflects the disease activity and MITAX was described by the International Myositis Assessment & Clinical Studies Group and (Isenberg et al., 2004). MITAX is widely accepted in the field and from Table S1 we can see that all patients from the Ctrl IIM group were in an active status (MITAX>0.14).

- **Line 301: the authors state that plasma IFN- α concentration was**

uniquely elevated in the MDA5+ DM-Act group compared with the Ctrl IIM and HD groups. However, FIG. 4G clearly shows that one and only one plasma is very largely above the others in terms of concentration. I'm not sure that the difference between the MDA5 group and the IIM control group is statistically significant without this plasma.

Response:

Thanks for the reviewer's professional comments and valuable suggestions. When we deleted the highest point of IFN- α in the MDA5+ DM group, the difference between the MDA5+ DM group and the Ctrl IIM group remained statistically significant (Please see the two figures below: left: original without deletion; right: after deleting the highest point in the MDA5+ DM group). Therefore, we prefer to keep the original figure in Fig. 4G.

- Line 304: The results of the IFN-gamma quantifications are interesting but difficult to understand, because the results presented in Fig.1E suggest a decrease in Th1 responses in the MDA5 group compared to the control groups. However, several studies have reported a higher production of IFN-gamma in MDA5 patients, which seems to correlate with the activity of the disease. This point should be discussed in depth by the authors, as it may result from a methodological limitation. From a global point of view, the whole discussion is centered around IFN-I, but

nothing is said about IFN-gamma.

Response :

We thank the reviewer's professional and valuable questions. We observed an increase of circulating IFN-gamma in the MDA5⁺ DM patients compared to the healthy donors, which is consistent with several early reports (Gono et al., 2014; Ishikawa et al., 2018). As we detected circulating IFN-gamma from the plasma and there are several cellular sources of IFN-gamma, including Th1, CD8⁺ T cell and NK cells. From our scRNA-seq data analysis, we did see a decreased proportion of scCD4T5-Th1, but we also observed increased proportions of scCD4T2-ISG, scCD8T4-GZMK+ GZMB+ Tm, scCD8T6-ISG, scCD8T7-pTm (Fig. 1E, 3B, and Fig. S3C-S3D). Furthermore, we confirmed an increased proportion of Ki67⁺ proliferating CD8⁺ T cells by flow cytometry (Fig. 3C), which indicates that CD8⁺ T cells could be an important source of IFN-gamma. We agree with the reviewer that due to the methodological limitation, we can not ascertain the major cellular source of the circulating IFN-gamma in this study, but we plan to address this issue in our future studies by directly measuring IFN-gamma-producing cells.

As suggested by the reviewer, we have added extensive discussion in the "Discussion" for the potential role of IFN-gamma in MDA5⁺DM, by including relevant literatures. "We also observed an increase of circulating IFN- γ in MDA5⁺ DM patients compared to HDs, which is consistent with several early reports (Gono et al., 2014; Ishikawa et al., 2018). A high level of IFN- γ is also implicated in the development and severity of MDA5⁺ DM. A positive correlation between serum IFN- γ levels and the computed tomography (CT) ground glass opacity (GGO) scores (G-scores) indicates that IFN- γ may play an important role in the lung pathophysiology of MDA5⁺ DM (Ishikawa et al., 2018). Furthermore, IFN- γ and TNF- α synergistically induce cytotoxicity in endothelial cells which may contribute to the unique cutaneous lesions and potentially the pulmonary lesions observed in MDA5⁺ DM (Yamaoka et al., 2002). IFN- γ also

synergizes with IL-1 \$\beta\$ to induce proinflammatory CX3CL1 in lung fibroblasts, which adds another potentially pathogenic role of IFN- \$\gamma\$ in MDA5⁺ DM (Isozaki et al., 2011)” (Line 513-523).

- Line 308: Again, the VEGF concentration is statistically different in the IIM group, but that's surely only based on the value of only one plasma....

Response :

We thank the reviewer's professional question. When we deleted the highest point of VEGF-D in the Ctrl IIM group, the statistical significance still existed between the MDA5⁺DM and the Ctrl IIM groups (Please see the two figures below: left: original without deletion; right: after deleting the highest point in the Ctrl IIM group). The concentrations of VEGF-D in Ctrl IIM group were significantly higher than those in the MDA5⁺DM group. So we prefer to keep the original figure in Fig. 4G.

Furthermore, it has been reported that VEGF family members were expressed in human skeletal muscle and the VEGF-A165b isoform was elevated in IIM patients (Volpi et al., 2013). The inflammation and coexisting atrophy/regeneration in IIM might trigger the upregulation of VEGF family members in muscle (Volpi et al., 2013). The roles of VEGF family members in IIM need investigations in future studies.

- Line 391: the authors state that ISG15⁺ CD8⁺ T cells are a promising

prognostic biomarker for MDA5+ DM patients, and they show higher frequencies of these cells in the DM-Act group. Once again, it is important to know the clinical activity of patients in the IIM group to know if this increased frequency of ISG15+ CD8+ T cells is specific to patients with MDA5+ DM, or if it simply reflects an active form, whatever the type of DM. In this part of the manuscript, the authors should also add anti-MDA5 antibody titers, since it is said that they have a predictive value (line 104 Introduction section).

Response:

We thank the reviewer's thoughtful advice. As suggested, we have added the disease activity score MITAX for both MDA5+DM and Ctrl IIM groups in the validation cohorts for B cells (Table S3) and T cells (Table S4). No significant difference of MITAX was observed between the MDA5+DM and Ctrl IIM groups for both validation cohorts. So we could conclude that the increased frequency of ISG15+ CD8+ T cells is preferentially increased in the patients with MDA5+DM compared to the Ctrl IIM patients.

We also added the data of anti-MDA5 titres in the prognostic analysis of the MDA5+DM patients. However, the anti-MDA5 titres from the patients at admission did not have a prognostic value for predicting survival in the univariate analysis (Fig. 7C and the table below). This result is quite understandable, as we only take the baseline anti-MDA5 titres to see its prognostic value. In the introduction, we have mentioned that "sustained high levels of anti-MDA5 are associated with unfavorable outcomes (Gono et al., 2012; Lian et al., 2020)" (Line 104 from the previous version). This is from a dynamic view to present an association between persistent high levels of anti-MDA5 titres and unfavorable outcomes in those refractory patients insensitive to treatment. To avoid potential misunderstanding, we have modified the sentence into "sustained high levels of anti-MDA5 are associated with unfavorable outcomes in those patients refractory to treatment (Gono et al., 2012; Lian et al., 2020)." (Line 102-103)

To further support the idea that anti-MDA5 alone may not a good marker to predict survival at early treatment stage of MDA5+DM. Nishioka et al. reported that the baseline serum anti-MDA5 titers did not differ significantly between patients who survived and those who succumbed to the disease (Nishioka et al., 2019). In another study, an initial decrease in anti-MDA5 titres after treatment was observed in most patients, regardless of survival or deceased, indicating that anti-MDA5 titre at admission was not adequate for predicting patient outcome (Abe et al., 2017). We have added relevant explanation in the revised manuscript: “It should be mentioned that the anti-MDA5 titres from the MDA5⁺ DM patients at baseline did not show a prognostic value in this study, which is consistent with the other two studies which also indicated that the baseline anti-MDA5 titres did not differentiate the patients who survived from those who succumbed to the disease (Abe et al., 2017; Nishioka et al., 2019).” (Line 408-412)

Characteristics	Univariate analysis			Multivariate analysis		
	HR	95%CI	P	HR	95%CI	P
Anti-MDA5 titre	0.811	0.438-1.501	0.505			
LDH [□]	1.001	1-1.003	0.057			
Ferritin	1.000	0.997-1.001	0.015	1.001	1.00-1.001	0.00271
MX1+%CD4 T	1.028	0.984-1.074	0.212			
ISG15+%CD4 T	1.017	0.993-1.041	0.173			
MX1+%CD8 T	1.032	1.001-1.063	0.041			
ISG15+%CD8 T	1.024	1.005-1.043	0.012	1.030	1.01-1.050	0.00315
MX1+%B	1.001	0.944-1.061	0.983			
ISG15+%B	1.013	0.981-1.045	0.431			

- Line 423: Tofacitinib also acts on the production of IFN-gamma, not only

on the type I IFN signaling pathway. Since the authors have demonstrated high concentrations of IFN-gamma in the plasma of MDA5+ DM patients, this point is also to be discussed.

Response:

We greatly appreciate the reviewer for the insightful suggestion. We agree with the reviewer that IFN-gamma signaling pathway is also inhibited by tofacitinib. We focused on the type I IFN signaling pathway in the manuscript mainly because upregulated type I IFN signature seemed to be more specific in MDA5+DM patients, as compared to type II IFN pathway. Nevertheless, IFN-gamma should also play an important role in the pathogenesis of myositis and could be a meaningful treatment target by JAKi tofacitinib.

As suggested by the reviewer, we have added the relevant content in the discussion of the revised manuscript: “As both type I and II IFNs activate Janus kinases, from a therapeutic point of view, the inclusion of Janus kinase inhibitors like tofacitinib in the treatment regimen will have a combined advantage to inhibit both the type I and II IFN signaling pathways.” (Line 523-526).

Reviewer #2 (Remarks to the Author):

The authors described in the present study the molecular features of MDA5+ DM patients by single-cell and BCR/TCR profiling.

They applied pretty standardized single-cell analysis methodologies; they were quite stringent in the criteria they used to select single cells for downstream analyses, which implies they have a high-quality dataset to work with. Reported statistical analysis is appropriate.

The limitation on the sample number, referred to in the discussion, is quite common for single cell studies, but the authors do not to make any overstatement about their findings, and are aware that further validations on other patient cohorts will be needed.

Overall, I find the manuscript well written, with a clear outline and well discussed.

Response:

We appreciate the reviewer's nice comments.

Reference:

Abe, Y., Matsushita, M., Tada, K., Yamaji, K., Takasaki, Y., and Tamura, N. (2017).

Clinical characteristics and change in the antibody titres of patients with anti-MDA5 antibody-positive inflammatory myositis. *Rheumatology (Oxford)* *56*, 1492-1497.

Gono, T., Kaneko, H., Kawaguchi, Y., Hanaoka, M., Kataoka, S., Kuwana, M., Takagi, K., Ichida, H., Katsumata, Y., Ota, Y., *et al.* (2014). Cytokine profiles in polymyositis and dermatomyositis complicated by rapidly progressive or chronic interstitial lung disease. *Rheumatology (Oxford)* *53*, 2196-2203.

Gono, T., Kawaguchi, Y., Kuwana, M., Sugiura, T., Furuya, T., Takagi, K., Ichida, H., Katsumata, Y., Hanaoka, M., Ota, Y., and Yamanaka, H. (2012). Brief report: Association of HLA-DRB1*0101/*0405 with susceptibility to anti-melanoma differentiation-associated gene 5 antibody-positive dermatomyositis in the Japanese population. *Arthritis Rheum* *64*, 3736-3740.

Isenberg, D.A., Allen, E., Farewell, V., Ehrenstein, M.R., Hanna, M.G., Lundberg, I.E., Oddis, C., Pilkington, C., Plotz, P., Scott, D., *et al.* (2004). International consensus outcome measures for patients with idiopathic inflammatory myopathies. Development and initial validation of myositis activity and damage indices in patients with adult onset disease. *Rheumatology (Oxford)* *43*, 49-54.

Ishikawa, Y., Iwata, S., Hanami, K., Nawata, A., Zhang, M., Yamagata, K., Hirata, S., Sakata, K., Todoroki, Y., Nakano, K., *et al.* (2018). Relevance of interferon-gamma in pathogenesis of life-threatening rapidly progressive interstitial lung disease in patients with dermatomyositis. *Arthritis Res Ther* *20*, 240.

Isozaki, T., Otsuka, K., Sato, M., Takahashi, R., Wakabayashi, K., Yajima, N., Miwa, Y., and Kasama, T. (2011). Synergistic induction of CX3CL1 by interleukin-1beta and interferon-gamma in human lung fibroblasts: involvement of signal transducer and activator of transcription 1 signaling pathways. *Transl Res* *157*, 64-70.

Lian, X., Zou, J., Guo, Q., Chen, S., Lu, L., Wang, R., Zhou, M., Fu, Q., Ye, Y., and Bao, C. (2020). Mortality Risk Prediction in Amyopathic Dermatomyositis Associated With Interstitial Lung Disease: The FLAIR Model. *Chest* *158*, 1535-1545.

Nishioka, A., Tsunoda, S., Abe, T., Yoshikawa, T., Takata, M., Kitano, M., Matsui, K., Nakashima, R., Hosono, Y., Ohmura, K., *et al.* (2019). Serum neopterin as well as ferritin, soluble interleukin-2 receptor, KL-6 and anti-MDA5 antibody titer provide markers of the response to therapy in patients with interstitial lung disease complicating anti-MDA5 antibody-positive dermatomyositis. *Mod Rheumatol* *29*, 814-820.

Volpi, N., Pecorelli, A., Lorenzoni, P., Di Lazzaro, F., Belmonte, G., Agliano, M., Cantarini, L., Giannini, F., Grasso, G., and Valacchi, G. (2013). Antiangiogenic VEGF isoform in inflammatory myopathies. *Mediators Inflamm* *2013*, 219313.

Yamaoka, J., Kabashima, K., Kawanishi, M., Toda, K., and Miyachi, Y. (2002). Cytotoxicity of IFN-gamma and TNF-alpha for vascular endothelial cell is mediated by nitric oxide. *Biochem Biophys Res Commun* *291*, 780-786.

REVIEWER COMMENTS

Reviewer #1 (Remarks to the Author):

The authors have perfectly addressed all the concerns in the revisions of the manuscript. Perhaps one last suggestion regarding the extension of discussion added in the revised manuscript, which addresses the levels of IFN-g observed in patients. In my opinion, it would also be interesting to briefly discuss the role of autoantibodies in the induction of IFNg, as suggested by the recent identification of autoAbs that directly stimulate IFNg production by PBMCs ("Monoclonal antibodies from B cells of patients with anti-MDA5 antibody -positive dermatomyositis directly stimulates interferon gamma production", J. Autoimmun, 2022 Jun;130:102831"). I let the authors decide whether to make this addition or not and I congratulate them again for their work.

Point-by-point Responses to the Reviewers' Comments

REVIEWER COMMENTS

Reviewer #1 (Remarks to the Author):

The authors have perfectly addressed all the concerns in the revisions of the manuscript. Perhaps one last suggestion regarding the extension of discussion added in the revised manuscript, which addresses the levels of IFN-g observed in patients. In my opinion, it would also be interesting to briefly discuss the role of autoantibodies in the induction of IFN γ , as suggested by the recent identification of autoAbs that directly stimulate IFN γ production by PBMCs ("Monoclonal antibodies from B cells of patients with anti-MDA5 antibody - positive dermatomyositis directly stimulates interferon gamma production", *J. Autoimmun.*, 2022 Jun;130:102831"). I let the authors decide whether to make this addition or not and I congratulate them again for their work.

Response:

We thank the reviewer's professional and valuable questions. We have added more discussion concerning the potential link between autoantibody and IFN- γ in MDA5⁺ DM patients: "The source of IFN- γ could be from multiple cell types, including T cells and NK cells. A recent study has indicated a role of autoantibodies from MDA5⁺ DM patients to induce IFN- γ secretion from peripheral mononuclear cells, thus making an interesting link between autoantibody response and IFN- γ production (Coutant et al., 2022)" (Line 673-680).

Reference:

Coutant, F., Bachet, R., Pin, J.J., Alonzo, M., and Miossec, P. (2022). Monoclonal antibodies from B cells of patients with anti-MDA5 antibody-positive dermatomyositis directly stimulate interferon gamma production. *J Autoimmun* 130, 102831.